# Adversarial robustness via robust low rank representations

**Pranjal Awasthi**
Google Research and Rutgers University.
pranjal.awasthi@rutgers.edu.

**Himanshu Jain**
Google Research.
himj@google.com.

**Ankit Singh Rawat**
Google Research.
ankitsrawat@google.com.

**Aravindan Vijayaraghavan**
Northwestern University.
aravindv@northwestern.edu.

## Abstract

Adversarial robustness measures the susceptibility of a classifier to imperceptible perturbations made to the inputs at test time. In this work we highlight the benefits of natural low rank representations that often exist for real data such as images, for training neural networks with certified robustness guarantees.

Our first contribution is for certified robustness to perturbations measured in $\ell_2$ norm. We exploit low rank data representations to provide improved guarantees over state-of-the-art randomized smoothing-based approaches on standard benchmark datasets such as CIFAR-10 and CIFAR-100.

Our second contribution is for the more challenging setting of certified robustness to perturbations measured in $\ell_\infty$ norm. We demonstrate empirically that natural low rank representations have inherent robustness properties, that can be leveraged to provide significantly better guarantees for certified robustness to $\ell_\infty$ perturbations in those representations. Our certificate of $\ell_\infty$ robustness relies on a natural quantity involving the $\infty \to 2$ matrix operator norm associated with the representation, to translate robustness guarantees from $\ell_2$ to $\ell_\infty$ perturbations. A key technical ingredient for our certification guarantees is a fast algorithm with provable guarantees based on the multiplicative weights update method to provide upper bounds on the above matrix norm. Our algorithmic guarantees improve upon the state of the art for this problem, and may be of independent interest.

## 1  Introduction

It is now well established across several domains like images, audio and natural language, that small input perturbations that are imperceptible to humans can fool deep neural networks at test time [1, 2, 3, 4]. This phenomenon known as *adversarial robustness* has led to flurry of research in recent years (see Section A for a discussion of related work). Following most prior work in this area [5, 6, 7, 8, 9], we will study the setting where adversarial perturbations to an input $x$ are measured in an $\ell_p$ norm ($p = 2$ or $p = \infty$).

In this work, we study methods for *certified adversarial robustness* in the framework developed in [10, 11]. The goal is to output a classifier $f$ that on input $x \in \mathbb{R}^n$ outputs a prediction $y$ in the label space $\mathcal{Y}$, along with a certified radius $r_f(x)$. The classifier is guaranteed to be robust at $x$ up to the radius $r_f(x)$ (with high probability), i.e., $\forall z : \|z\|_p \le r_f(x), f(x + z) = f(x)$.

For an $\ell_p$ norm and $\varepsilon > 0$, the certified accuracy of a classifier $f$ is defined as

$$\mathrm{acc}_\varepsilon^{(\ell_p)}(f) = \mathop{\mathbb{P}}_{(x,y)\sim D}\big[f(x) = y \text{ and } r_f(x) \geq \varepsilon\big], \tag{1}$$

where $D$ is the data distribution generating test inputs. We call the radius $r_f(x)$ returned by the classifier as the *certified radius* on $x$. When $\varepsilon = 0$ this is the natural accuracy of $f$.

For certified adversarial robustness to $\ell_2$ perturbations, the *randomized smoothing* procedure proposed in [10, 11] is a simple and efficient method that can be applied to *any* neural network. Randomized smoothing works by creating a smoothed version of a given classifier by adding Gaussian noise to the inputs (see Section 2). The smoothed classifier exhibits certain Lipschitzness properties, and one can derive good certified robustness guarantees from it. The study of randomized smoothing for certified $\ell_2$ robustness is an active research area and the current best guarantees are obtained by incorporating the smoothed classifier into the training process [12] (see Section A).

It seems much more challenging to obtain certified adversarial robustness to $\ell_\infty$ perturbations [13, 14, 15]. In particular, the design of a procedure akin to randomized smoothing has been difficult to achieve for $\ell_\infty$ perturbations. One approach to obtain certified $\ell_\infty$ robustness is to translate a certified radius guarantee of $\varepsilon$ for $\ell_2$ perturbations (via randomized smoothing) into an $\varepsilon/\sqrt{n}$ certified radius guarantee for $\ell_\infty$ perturbations; here $n$ is the dimensionality of the ambient space. Furthermore, recent work [16, 17, 18] has established lower bounds proving that randomized smoothing based methods cannot break the above $\sqrt{n}$ barrier for $\ell_\infty$ robustness in the worst case.

However real data such as images are not worst case and often exhibits a natural low rank structure. In this work we show how we can leverage such natural low-rank representations for the data, in order to design algorithms based on randomized smoothing with improved certified robustness guarantees for both $\ell_2$ and $\ell_\infty$ perturbations.

**Our Contributions.** We now describe our main contributions.

*Improved certified $\ell_2$ robustness:* Our first contribution is to design new smoothed classifiers for achieving certified robustness to $\ell_2$ perturbations. These classifiers achieve improved tradeoffs between natural accuracy and certified accuracy at higher radii. We achieve this by leveraging the existence of good low-rank representation for the data. We modify the randomized smoothing approach to instead selectively inject more noise along certain directions, without compromising the accuracy of the classifier. The large amount of noise leads to classifier that is less sensitive to $\ell_2$ perturbations, and hence achieves higher certified accuracy across a wide range of radii. We empirically demonstrate the improvements obtained by our approach on image data in Section 2.

*Fast algorithms for translating certified robustness guarantees from $\ell_2$ to $\ell_\infty$:* For the more challenging setting of $\ell_\infty$ robustness we consider classifiers of the form $f(Px)$ where $P$ is an arbitrary linear map, and $f$ represents an arbitrary neural network. When translating certified robustness guarantees for $\ell_2$ perturbations to obtain guarantees for $\ell_\infty$ perturbations, the loss incurred is captured by the $\infty \to 2$ operator norm of matrix $P$. While computing this operator norm is NP-hard, we design a fast approximate algorithm based on the multiplicative weights update method with provable guarantees. Our algorithmic guarantees give significant improvements over the best known bounds [19, 20] for this problem and may be of independent interest (see Section 3).

*Certified $\ell_\infty$ robustness in natural data representations:* Real data such as images have natural representations that are often used in image processing e.g., via the Discrete Cosine Transform (DCT). Via an empirical study we highlight the need for achieving $\ell_\infty$ robustness in the DCT basis. More importantly, we demonstrate that the representation in the DCT basis is robust, i.e., there exist low rank projections that capture most of the signal in the data and that at the same time have small $\infty \to 2$ operator norm.[1] We develop a fast heuristic based on sparse PCA to find such robust projections. When combined with our

multiplicative weights based algorithm, this leads to a new training procedure based on randomized smoothing. Our procedure can be applied to any network architecture and provides stronger guarantees on robustness to $\ell_\infty$ perturbations in the DCT basis.

## 2 Certified Robustness to $\ell_2$ Perturbations

We build upon the *randomized smoothing* technique proposed in [10, 11] and further developed in [12]. Consider a multiclass classification problem and a classifier $f : \mathbb{R}^n \to \mathcal{Y}$, where $\mathcal{Y}$ is the label set. Given $f$, randomized smoothing produces a smoothed classifier $g$ where

$$g(x) = \arg\max_{y \in \mathcal{Y}} \mathbb{P}(f(x + \delta) = y). \tag{2}$$

Here $\delta \sim N(0, \sigma^2 I)$ is the Gaussian noise added. The following proposition holds.

**Proposition 2.1** ([10, 11])**.** *Given a classifier $f$, let $g$ be its smoothed version as defined in* (2) *above. On an input $x$, and for $\delta \sim N(0, \sigma^2 I)$ define $y_A \coloneqq \arg\max_y \mathbb{P}(f(x + \delta) = y)$, and let $p_A = \mathbb{P}(f(x + \delta) = y_A)$. Then the prediction of $g$ at $x$ is unchanged up to $\ell_2$ perturbations of radius*

$$r(x) = \frac{\sigma}{2}\big(\Phi^{-1}(p_A) - \Phi^{-1}(p_B)\big). \tag{3}$$

*Here $p_B = \max_{y \neq y_A} \mathbb{P}(f(x + \delta) = y)$ and $\Phi^{-1}$ is the inverse of the standard Gaussian CDF.*

Hence, randomized smoothing provides a fast method to certify the robustness of any given classifier on various inputs. In order to get robustness to large perturbations it is desirable to choose the noise magnitude $\sigma$ as large as possible. However, there is a natural tradeoff between the amount of noise added and the natural accuracy of the classifier. As an example consider an input $x \in \mathbb{R}^n$ of $\ell_2$ length $\sqrt{n}$. If $\sigma$ is the average amount of noise added then one is restricted to choosing $\sigma$ to be a small constant in order for the noise to not overwhelm the signal.

However, it is well known that natural data such as images are low dimensional in nature. Figure 5 in Appendix B shows that for the CIFAR-10 and CIFAR-100 datasets, even when projected, via PCA, onto 200 dimensions, the reconstruction error remains small. If the input is close to an $r$-dimensional subspace, then it is natural to add noise only within the subspace for smoothing. Formally, let $\Pi$ be the projection matrix on to an $r$-dimensional subspace and $x$ be such that $\|\Pi x\|_2 \approx \|x\|_2 = \sqrt{n}$. For $\delta \sim N(0, \sigma^2 I)$ we have $\|\Pi \delta\|_2 \approx \sigma\sqrt{r}$. Hence if we only add noise within the subspace, then $\sigma$ can be as large as $\sqrt{n/r}$ as opposed to a constant without significantly affecting the natural accuracy.

We formalize this into an efficient training algorithm as follows: we take a base classifier/neural network $f(x)$ and replace it with the smoothed classifier $g_\Pi(x)$ where

$$g_\Pi(x) = \arg\max_{y \in \mathcal{Y}} \mathbb{P}(f(\Pi x + \delta_\Pi) = y). \tag{4}$$

where $\Pi$ is a projection matrix onto an $r$-dimensional subspace and $\delta_\Pi$ is a standard Gaussian of variance $\sigma^2$ that lies within $\Pi$. For data such as images, good projections $\Pi$ can be obtained via methods like PCA. Furthermore, certifying the robustness of our proposed smoothed classifier can be easily incorporated into existing pipelines for adversarial training with minimal overhead. In particular using the rotational symmetry of Gaussian distributions it is easy to show the following

**Proposition 2.2.** *Given a base classifier $f : \mathbb{R}^n \to \mathcal{Y}$ and a projection matrix $\Pi$, on any input $x$, the smoothed classifier $g_\Pi(x)$ as defined in* (4) *is equivalent to the classifier given by*

$$\tilde{g}_\Pi(x) = \arg\max_{y \in \mathcal{Y}} \mathbb{P}(f\big(\Pi(x + \delta)\big) = y). \tag{5}$$

*Here $\delta$ is a standard Gaussian of variance $\sigma^2$ in every direction.*

Hence constructing our proposed smoothed classifier simply requires adding a linear transformation layer to any existing network architecture before training via randomized smoothing. We propose to train the smoothed classifier as defined in (5) by minimizing its adversarial

standard cross entropy loss as proposed in [6]. However, since dealing with $\arg\max$ is hard from an optimization point of view, we follow the approach of [12] and instead minimize the cross entropy loss of the following soft classifier

$$G_\Pi(x) = \mathop{\mathbb{E}}_{\delta \sim N(0,\sigma^2 I)}[f(\Pi(x+\delta))]. \tag{6}$$

This leads to the following objective where $\ell_{ce}$ is the standard cross-entropy objective and $\varepsilon > 0$ is perturbation radius chosen for the training procedure.

$$\arg\min_f \mathop{\mathbb{E}}_{(x,y)} \Big[ \max_{z:\|z\|_2 \le \varepsilon} \ell_{ce}(G_\Pi(x+z), y)\Big]. \tag{7}$$

Following [6, 12], the inner maximization of finding adversarial perturbations is solved via projected gradient descent (PGD), and given the adversarial perturbations, the outer minimization uses stochastic gradient descent. This leads to the following training procedure.

---

**Algorithm 1** Adversarial training via projections

---

1: **function** ROBUSTTRAIN(training data $(x_1, y_1), \ldots, (x_m, y_m)$, subspace rank $r$, base noise magnitude $\sigma$, $\lambda \in [0, 1]$, number of steps $T$, mini batch size $b$)
2:     Perform PCA on (unlabeled) data matrix $A \in \mathbb{R}^{n \times m}$ to obtain a rank-$r$ projection matrix $\Pi$.
3:     Set $G_\Pi$ as in (6) with $\sigma = \lambda\sqrt{n/r}$.
4:     **for** $t = 1, \ldots, T$ **do**
5:         Obtain a mini batch of $b$ examples $(x_{t_1}, y_{t_1}), \ldots, (x_{t_b}, y_{t_b})$.
6:         For each $x_{t_i}$ use projected gradient ascent on inner maximization in (7) to get $x'_{t_i}$.
7:         Given perturbed examples $\{(x'_{t_i}, y_{t_i})\}_{i \in [b]}$, update network parameters via SGD.
8:     Output the smoothed classifier $\tilde{g}_\Pi(x)$.

---

**Empirical Evaluation.** We compare Algorithm 1 with the algorithm of [12] for various values of $\sigma$ and $\varepsilon$ (used for training to optimize (7)). We choose $\varepsilon \in \{0.25, 0.5, 0.75, 1.0\}$ and for each $\varepsilon$ we choose the value of $\sigma$ as described in [12]. In each case, we train the classifier proposed in [12] using a noise magnitude $\sigma$, and we train our proposed smoothed classifier using higher noise values of $\lambda\sigma\sqrt{n/r}$, where $\lambda$ is a parameter that we vary. To obtain the projection matrix $\Pi$ we perform a PCA onto each image channel separately and use the top 200 principal components to obtain the projection matrix $\Pi$.

In all experiments, we train a ResNet-32 network on the CIFAR-10 dataset by optimizing (7). The complexity of Algorithm 1 is comparable to the complexity of training a smoothed classifier as in the work of [SYL+ 21 19]. The PCA step incurs a one time preprocessing cost and the projection step at the beginning simply corresponds to adding a linear layer to an existing ResNet architecture. As an example, on the CIFAR-10 dataset, for $\varepsilon = 0.25$, training the classifier of [12] takes on average 21.27 seconds per epoch, whereas Algorithm 1 takes 21.29 seconds per epoch on average. The same behavior holds across different parameter settings. Figure 1 shows a comparison of certified accuracies for different radii and different values of $\lambda$. See Appendix B for a description of the hyperparameters and additional experiments. For application to other domains such as text data where the input representation is not fixed, training the linear projection $\Pi$ along with the network could be beneficial and in fact necessary. For image datasets above, we also experimented with simultaneously training the projection with the network parameters. The results obtained were similar to using a fixed projection and we did not see any significant advantage.

As can be seen from Figure 1, varying the value of $\lambda$ lets us tradeoff lower accuracy at small values of the radius for a significant gain in certified accuracy at higher radii as compared to the method of [12]. In particular we find that choosing values of $\lambda$ close to 0.5 leads to networks that can certify accuracy at much higher radii with minimal to no loss in the natural accuracy as compared to the approach of [12].

In Figure 2 we present the result of our training procedure for various values of $\varepsilon$ and $\sigma$ and compare with the $\ell_2$ smoothing method of [12] on the CIFAR-10 and CIFAR-100 datasets. For both datasets, our trained networks outperform the method of [12] across a large range of radius values. For higher values of radius (say, $\gtrsim 0.5$) our method achieves a desired certified

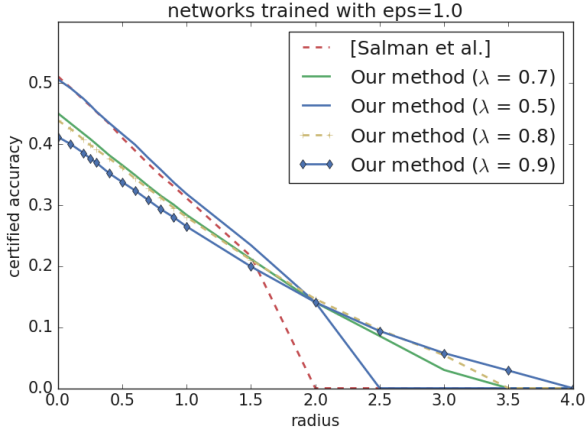

Figure 1: Plot of certified accuracy achieved at different radii when different values of $\lambda$ are chosen to optimize the smoothed classifier in (5). We compare with the method of [12] The plot is obtained by training a ResNet-32 architecture on CIFAR-10 with $\varepsilon = 1.0$. The $y$-intercept of each curve represents the natural accuracy of the corresponding classifier.

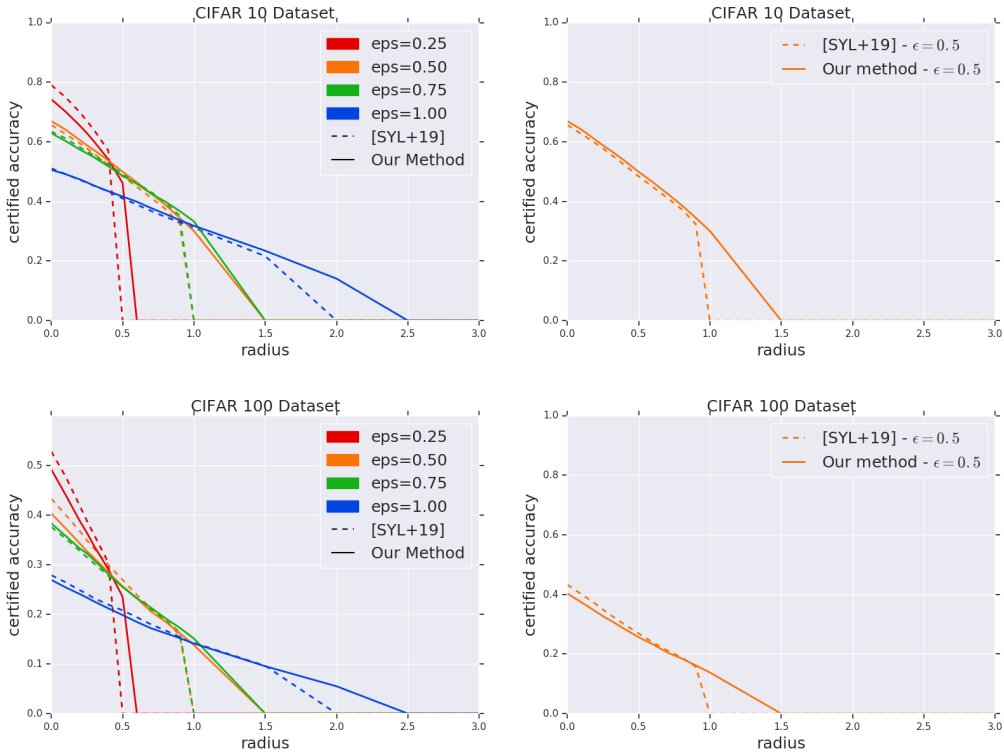

Figure 2: A comparison of certified radius guarantees obtained via Algorithm 1 as compared to the approach of [12]. The x-axis is the radius, and the y-axis represents the certified accuracy. The top row describe results for the CIFAR-10 dataset – (left) certified accuracies for various values of $\varepsilon$, (right) for $\varepsilon = 0.5$. Similarly, the bottom row describe the results for the CIFAR-100 dataset.

accuracy with significantly higher natural accuracy as compared to the method of [12]. For instance in the CIFAR-10 dataset, at a radius of 1.0 and a desired certified accuracy of at least 0.35, the method of [12] achieves a natural accuracy of $\approx 0.5$ (blue dotted curve at radius 0). In contrast our method achieves the same with a natural accuracy of $\approx 0.65$ (green solid curve at radius 0). On the other hand, at very small radius values the method of [12] is better. This is expected as we suffer a small loss in natural accuracy due to the PCA step in Algorithm 1. We remark that in practice, we may not know the radius of adversarial perturbation (and the ideal choice of $\varepsilon$) beforehand. Hence sacrificing a small amount of accuracy at small radii for a significant gain at higher radii is a desirable tradeoff.

# 3 Methods for Certified $\ell_\infty$ Robustness

We now describe our algorithms for the more challenging problem of certified robustness to $\ell_\infty$ perturbations in a given basis or representation. Our approach is to leverage the existence of good representations of natural data measured by a certain robustness parameter, and translate $\ell_2$ robustness guarantees from Section 2 to get certified $\ell_\infty$ guarantees. Consider $f : \mathbb{R}^n \to \mathcal{Y}$ and $g(x) := f(\Pi x)$, where $\Pi \in \mathbb{R}^{n \times n}$ represents a projection matrix. The certified accuracy of $g$ satisfies

$$\forall \varepsilon > 0, \ \ \mathrm{acc}_{\varepsilon'}^{(\ell_\infty)}(g) \geq \mathrm{acc}_\varepsilon^{(\ell_2)}(g) \text{ for } 0 \leq \varepsilon' \leq \varepsilon/\|\Pi\|_{\infty \to 2}, \ \ \text{where } \|\Pi\|_{\infty \to 2} = \max_{x : \|x\|_\infty \leq 1} \|\Pi x\|_2$$

is the $\infty \to 2$ operator norm of $\Pi$ and represents a robustness parameter (see Proposition C.2 for a formal claim). Hence, to translate guarantees from $\ell_2$ to $\ell_\infty$, we look for *robust* projections $\Pi$ that have small $\infty \to 2$ operator norm. Our approach is inspired by the recent theoretical work of [21], and finds a low-dimensional representation given by an (orthogonal) projection matrix $\Pi$ for the data that has small $\|\Pi\|_{\infty \to 2}$. By matrix norm duality, there is a nice characterization for $\|\Pi\|_{\infty \to 2}$ as the maximum $\ell_1$ norm among Euclidean unit vectors in the subspace of $\Pi$ (this is a notion of sparsity of the vectors). For a rank $r$ projector $\Pi$, the range of values taken by $\|\Pi\|_{\infty \to 2}$ is $[\sqrt{r}, \sqrt{n}]$. Hence if the projection is not low-rank and sparse, $\|\Pi\|_{\infty \to 2}$ could be as large as $\sqrt{n}$ (e.g., when $\Pi = I$). This is consistent with the loss of $\sqrt{n}$ factor in robustness radius to $\ell_\infty$ perturbations for general datasets [16, 17]. Moreover as we have seen in Section 2, good low-rank representations of the data also give stronger certified $\ell_2$ robustness guarantees (and in turn, stronger certified $\ell_\infty$ guarantees).

The goal is to find a good robust rank-$r$ projection of the data if it exists. We propose a heuristic based on sparse PCA [22] to find a robust projection with low error (see Section D.2). Since we aim for certified robustness, an important step is to compute and certify an upper bound on $\|\Pi\|_{\infty \to 2}$, for a projection $\Pi$. This is an NP-hard problem, related to computing the famous Grothendieck norm [23]. We describe a new, scalable, approximate algorithm for computing upper bounds on $\|\Pi\|_{\infty \to 2}$ with provable guarantees.

**Certifying the $\infty \to 2$ operator norm.** Our fast algorithm is based on the multiplicative weights update (MWU) method for approximately solving a natural semi-definite programming (SDP) relaxation, and produce a good upper bound on $\|\Pi\|_{\infty \to 2}$. Our upper bound also comes with a certificate from the dual SDP i.e., a short proof of the correctness of the upper bound. Given a candidate $\Pi$ our algorithm will compute an upper bound for $\|\Pi\|_{\infty \to 1}$ which by matrix norm duality satisfies

$$\|\Pi\|_{\infty \to 2}^2 = \|\Pi\|_{\infty \to 1} = \max_x x^\top \Pi x \ \text{ subject to } \|x\|_\infty \leq 1. \tag{8}$$

This problem falls into the more general class of problems called Quadratic Programming:

$$\text{Given a symmetric matrix } M \text{ with } \forall i \in [n] : \ M_{ii} \geq 0, \ \ \max_{x : \|x\|_\infty \leq 1} x^\top M x. \tag{9}$$

The standard SDP relaxation for the problem (see (12) in Appendix D.1), has primal variables represented by the positive semi-definite (PSD) matrix $X \in \mathbb{R}^{n \times n}$ satisfying constraints $X_{ii} \leq 1$ for each $i \in [n]$. The SDP dual of this relaxation (given in (14) of Appendix D.1) has variables $y_1, \ldots, y_n \geq 0$ corresponding to the $n$ constraints in the primal SDP. Since the SDP is a valid relaxation for (9), it provides an upper bound for $\infty \to 1$ operator norm[2]. Classical results show that it is always within a factor of $\pi/2$ of the actual value of $\|M\|_{\infty \to 1}$ [24, 25]. However, it is computationally intensive to solve the SDP using off-the-shelf SDP solvers (even for CIFAR-10 images, $X$ is $1024 \times 1024$). We design a fast algorithm based on the *multiplicative weight update* (MWU) framework [26, 19].

**Description of the algorithm** Our algorithm differs slightly from the standard MWU approach for solving the above SDP. The algorithm below takes as input a matrix $M$ and always returns a valid upper bound $\mathrm{upbd}_{\min}$ on the SDP value, along with a dual feasible

solution $y_{\text{ubmin}}$ that can act as a certificate, and a candidate primal solution $X$ that attains the same value (and is potentially feasible). Theorem 3.1 proves that for the right setting of parameters, particularly the number of iterations $T_f = O(n \log n/\delta^3)$, the solution $X$ is also guaranteed to be feasible up to small error $\delta > 0$.

---

**Algorithm 2** Fast Certification of $\infty \to 1$ norm and Quadratic Programming

---

1: **function** CERTIFYSDP($M \in \mathbb{R}^{n \times n}$, iteration bound $T_f$, slack $\delta$, damping $\rho$)
2:     Initialize $\alpha = (1, 1, \ldots, 1) \in \mathbb{R}^n$. primal $X = 0$, dual $y = 0^n$, upbd$_{\min} = \infty$, $y_{\text{ubmin}} = 0^n$.
3:     **for** $t = 0, 1 \ldots, T$ **do**
4:         $\tilde{\alpha} \leftarrow (1 - \delta)\alpha + \delta(1, 1, \ldots, 1)$.
5:         $\lambda \leftarrow$ max-eigenvalue($\text{diag}(\tilde{\alpha})^{-1/2} M \text{diag}(\tilde{\alpha})^{-1/2}$) and $u \in \mathbb{R}^n$ be its eigenvector.
6:         $v \leftarrow \sqrt{n} \cdot \text{diag}(\tilde{\alpha})^{-1/2} u$, $y \leftarrow \frac{1}{t+1}(ty + \lambda \tilde{\alpha})$, $X \leftarrow \frac{1}{t+1}(tX + vv^\top)$.
7:         **if** $\|v\|_\infty \leq 1 + \delta$ or $\max_i X_{ii} \leq 1 + \delta$ **then**, do early stop and return appropriate values.
8:         Update $\forall i \in [n], \alpha(i) \leftarrow \alpha(i) \exp\left(\frac{\delta}{\rho}(v(i)^2 - 1)\right)$, and renormalize s.t. $\sum_{i=1}^n \alpha(i) = n$.
9:         **if** upbd$_{\min} > n\lambda$, **then** set upbd$_{\min} = n\lambda$ and $y_{\text{ubmin}} = \lambda\tilde{\alpha}$.
10:     Output upbd$_{\min}$, dual solution $y_{\text{ubmin}}$, and primal candidate $X$.

---

Recall that from (8) an estimate of the the $\infty \to 1$ norm immediately translates to an estimate of the $\infty \to 2$ norm. In the above algorithm, there are weights given by $\alpha$ for $n$ different constraints of the form $X_{ii} \leq 1$. At each iteration, the algorithm maximizes the objective subject to *one* constraint of the form $\sum_i \tilde{\alpha}_i X_{ii} \leq n$, where $\tilde{\alpha}$ involves a small correction to $\alpha$ that is crucial to ensure the run-time guarantees. The maximization is done using a maximum eigenvalue/eigenvector computation. The weights $\alpha$ are then updated using a multiplicative update based on the violation of the solution found in the current iterate. The damping factor $\rho$ determines the rate of progress of the iterations – the smaller the value of $\rho$ the faster the progress, but a very small value may lead to oscillations. A more aggressive choice of $\rho$ compared to the one in Theorem 3.1 seems to work well in practice. Finally, we remark that for *every* choice of $\alpha$ and $\rho$ we get a valid upper bound (due to dual feasibility). We show the following guarantee for our algorithm for problem (9).

**Theorem 3.1.** *Suppose $\delta > 0$, and $M$ be any symmetric matrix with $M_{ii} \geq 0$ $\forall i \in [n]$. For any $\alpha \in \mathbb{R}_{\geq 0}^n$ with $\sum_{i=1}^n \alpha(i) = n$, if $\lambda = \lambda_{\max}\left((diag(\alpha)^{-1/2} M diag(\alpha)^{-1/2})\right)$, then $y = \lambda\alpha$ is feasible for the dual SDP and gives a valid upper bound of $n\lambda$ on the objective value for the SDP relaxation to (9). Moreover Algorithm 2 on input $M$, with parameters $\delta$ and $\rho = O(n/\delta)$ after $T = O(n \log n/\delta^3)$ iterations finds a feasible SDP solution $\widehat{X} \succeq 0$ and a feasible dual solution $\widehat{y} \in \mathbb{R}^n$ that both sandwich the optimal SDP value within a $1 + \delta$ factor.*

(See Prop D.1 and Theorem D.2 in Appendix D.1 for formal statements along with proofs. ) Each iteration only involves a single maximum eigenvalue computation, which can be done up to $(1 + \delta)$ accuracy in $T_{eig} = \tilde{O}(m/\delta)$ time where $m$ is the number of non-zeros in $M$ (see e.g., [26]). To the best of our knowledge, this gives significant improvements over the prior best bound of $\tilde{O}(n^{1.5}m/\delta^{2.5})$ runtime for solving the above SDP [19, 20]. Our algorithm and analysis differs from the general MWU framework [20] by treating the objective differently from the constraints so that the "width parameter" does not depend on the objective. A crucial step in our algorithm and proof is to add a correction term of $O(\delta)$ to the weights in each step that ensures that the potential violation of each constraint in an iteration is bounded. In addition our algorithm is more scalable than existing off-the-shelf methods. See Appendix D.1 and F for details and comparisons.

## 4   Training Certified $\ell_\infty$ Robust Networks in Natural Representations.

Building upon our theoretical results from the previous section we now demonstrate that for natural representations, one can indeed achieve better certified robustness to $\ell_\infty$ perturbations

by translating guarantees from certified $\ell_2$ robustness. We focus on image data, and study the representation of images in the DCT basis. Before we describe the details of our training and certification procedure for $\ell_\infty$ robustness, we provide further empirical evidence that imperceptibility in natural representations such as the DCT basis is a desirable property.

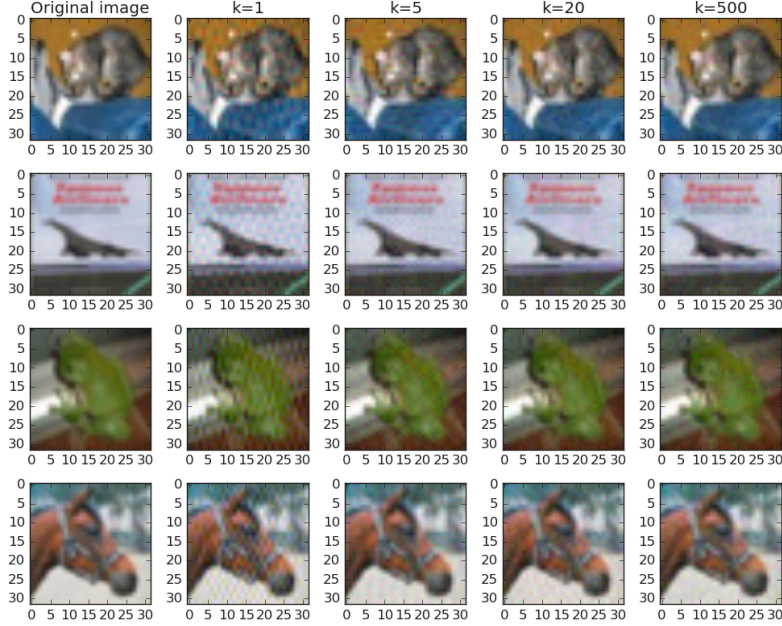

Figure 3: Original images (leftmost) from the CIFAR-10 dataset and their perturbed versions when sparse random perturbations are added in the DCT basis with sparsity $k$. Large perturbations in the DCT basis (e.g., $k = 1$ ) lead to perceptible changes in the pixel space though they are $\ell_\infty$ perturbations of $\varepsilon \le 0.09$. As $k$ increases the imperceptibility of the perturbed images improves.

**Study of imperceptibility in DCT basis.** We argue that for adversarial perturbations to be imperceptible to humans they should be of small magnitude in the DCT representation (perhaps in addition to being small in the pixel basis). We take images from the CIFAR-10 dataset in its DCT representation and add sparse random perturbations to them. In particular, for a sparsity parameter $k$, we pick $k$ coordinates in the DCT basis at random and add a random perturbation with $\ell_\infty$ norm of $c_k/\sqrt{k}$ where $c_k \approx \varepsilon\sqrt{n}$ is chosen such that the perturbed images are $\varepsilon \le 0.09$ away from the unperturbed images in the pixel space. Notice that for small values of $k$, a perturbation of large $\ell_\infty$ norm is added. Figure 3 visualizes the perturbed images for different values of $k$. As seen, large perturbations in the DCT basis lead to visually perceptible changes, even if they are ($\varepsilon \le 0.09$)-close in the pixel basis. For comparison we also include in Appendix E imperceptible adversarial examples for these images that were generated via the PGD based method of [6] on a ResNet-32 network trained on the CIFAR-10 dataset for robustness to $\ell_\infty$ perturbations of magnitude $\varepsilon = 0.09$. This further motivates studying robustness in natural data representations.

**Training certified $\ell_\infty$ robust networks in the DCT basis.** The methods developed in Section 3 and 2 together give algorithms for training classifiers with certified $\ell_\infty$ robustness. However while we want $\ell_\infty$ robustness in a different representation $\mathcal{U}$ (e.g., in the DCT basis), it may still be more convenient to use off-the-shelf methods for performing the training in the original representation $\mathcal{X} \subseteq \mathbb{R}^n$ (e.g., pixel representation). Let the orthogonal matrix $O \in \mathbb{R}^{n \times n}$ represent the DCT transformation. Consider an input $x \in \mathcal{X}$ and let $u = Ox \in \mathcal{U}$ be its DCT representation, where $\mathcal{U}$ is the space of images in the DCT basis. It is easy to see that functions $f : \mathcal{X} \to \mathcal{Y}$ and $g : \mathcal{U} \to \mathcal{Y}$ given by $g(u) := f(O^{-1}u) = f(x)$ have the same certified accuracy to $\ell_2$ perturbations. Moreover if the classifier $g$ satisfies $g(u) = g(\Pi u)$ for some projection $\Pi$, the robust accuracy of $g$ to $\ell_\infty$ perturbations in the representation $\mathcal{U}$

satisfies (see Proposition C.3 in appendix)

$$\operatorname{acc}_{\varepsilon'}^{(\ell_\infty)}(g) \geq \operatorname{acc}_{\varepsilon}^{(\ell_2)}(f), \text{ for any } \varepsilon > 0, 0 \leq \varepsilon' \leq \varepsilon/\|\Pi\|_{\infty\to 2}. \tag{10}$$

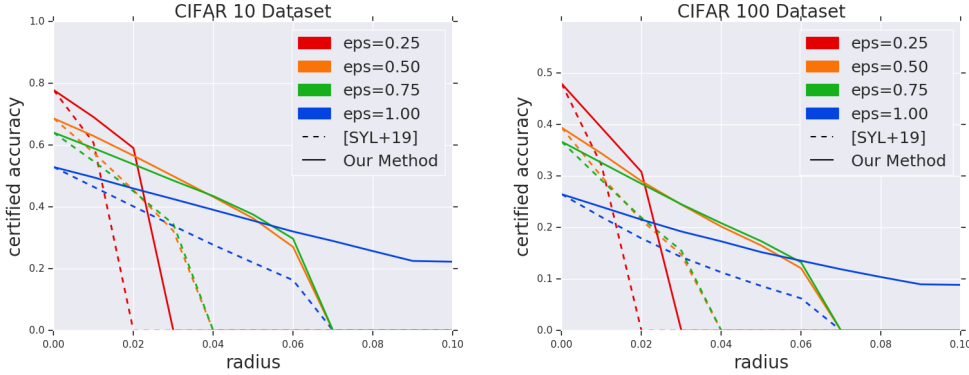

Figure 4: A comparison of certified $\ell_\infty$ accuracy (y-axis) in the DCT basis of our method to that of [12], for different values of $\varepsilon$ and with different certified radii on the x-axis, for $\lambda = 0.5$. The left and right plots describe results for CIFAR-10 and CIFAR-100 datasets.

**Experimental Data.** We evaluate our approach on the CIFAR-10 and CIFAR-100 datasets. From (10), it is sufficient to train a classifier in the original pixel space with an appropriate projection $\Pi' = O\Pi$. Hence, we train a smoothed classifier as defined in (2) using Algorithm 1. To obtain the required $\Pi$, we first use the sparse PCA based heuristic (Algorithm 3) to find a projection matrix of rank 200 for the three image channels separately. We then use Algorithm 2 to compute upper bounds on the $\infty \to 2$ operator norm of the projections matrices. Finally, we combine the obtained projection matrices from each channel to obtain a projection $\Pi$. Table 1 shows the values of the operator norms certified by our algorithm for each image channel and for the combined projection matrix. Notice the obtained subspaces have operator norm values significantly smaller than $\sqrt{n} = 55.42$. The reconstruction error in each case, when projected onto $\Pi$ is at most 0.0345.

After training, on an input $x$ we obtain a certified radius for $\ell_\infty$ perturbations in the DCT basis by obtaining a certified $\ell_2$ radius via randomized smoothing and then dividing the obtained value by $\|\Pi\|_{\infty\to 2}$. We compare with the approach of [12] for training a smoothed classifier without projections. Since the classifier of [12] does not involve projections, we translate the resulting $\ell_2$

| Dataset | R | G | B | $\Pi$ |
|---------|-------|-------|-------|-------|
| CIFAR-10 | 17.45 | 17.51 | 17.39 | 30.22 |
| CIFAR-100 | 17.22 | 17.33 | 17.37 | 29.97 |

Table 1: The table shows bounds on $\infty \to 2$ norm for projection matrices obtained by Algorithm 2 on CIFAR-10 and CIFAR-100 training sets.

robustness guarantee into an $\ell_\infty$ guarantee by dividing with $\sqrt{n} = 55.42$ as done in [12]. Figure 4 shows that across a range of training parameters, our proposed approach leads to significantly higher certified accuracy to $\ell_\infty$ perturbations in the DCT basis.

## 5 Conclusion

In this paper, we have shown significant benefits in leveraging natural structure that exists in real-world data e.g., low-rank or sparse representations, for obtaining certified robustness guarantees under both $\ell_2$ perturbations and $\ell_\infty$ perturbations in natural data representations. Our experiments involving imperceptibility in the DCT basis for images suggest that further study of $\ell_\infty$ robustness for other natural basis (apart from the co-ordinate basis) would be useful for different data domains like images, audio etc. We also gave faster algorithms for approximately solving semi-definite programs for quadratic programming (with provable guarantees that improve the state-of-the-art), to obtain certified $\ell_\infty$ robustness guarantees. Such problem-specific fast approximate algorithms for powerful algorithmic techniques like SDPs and other convex relaxations may lead to more scalable certification procedures with better guarantees. Finally it would be interesting to see if our ideas and techniques involving the $\infty \to 2$ operator norm can be adapted into the training phase, in order to achieve better certified $\ell_\infty$ robustness in any desired basis without compromising much on natural accuracy.

## Broader Impact

Our work provides efficient algorithms for training neural networks with certified robustness guarantees. This can have significant positive societal impact considering the importance of protecting AI systems against malicious adversaries. A classifier with certified robustness guarantees can give a sense of security to the end user. On the other hand, our methods achieve robustness at the expense of a small loss in natural test accuracy as compared to non-adversarial training. It is unclear how this loss in accuracy is distributed across the population. This could have a negative societal impact if the loss in accuracy is disproportionately on data points/individuals belonging to a specific demographic group based on say race or gender. That said, robustness to perturbations also corresponds to a natural notion of individual fairness since data points with similar features need to be treated similarly by a robust classifier. Hence, a careful study must be done to understand these effects before a large scale practical deployment of systems based on our work.

## Acknowledgments and Disclosure of Funding

The first author acknowledges support from the NSF HDR TRIPODS award CCF-1934924. The last author was supported by the National Science Foundation (NSF) under Grant No. CCF-1652491, CCF-1637585 and CCF-1934931.

## Footnotes

[1]This is also true for domains such as audio in the DCT basis. See Appendix G for experimental evidence.

[2]A fast algorithm that potentially finds a local optimum for the problem will not suffice for our purposes; we need an upper bound on the global optimum.

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
