[Supplementary Material]

# A    Related Work

Several recent works have aimed to design algorithms that are robust to adversarial perturbations. This is a rapidly growing research area, and here we survey the works most relevant in the context of the current paper.

In practice, first order methods are a popular choice for adversarial training. Algorithms such as the *fast gradient sign method* (FGSM) [5] and *projected gradient descent* (PGD) [6] fall into this category. These methods are appealing since they are generally applicable to many network architecture and only rely on black box access to gradient information. Recent works have aimed to improve upon the scalability and runtime efficiency of the above approaches [7, 8, 9, 27, 28]. In particular, [27, 28] study the use of low-rank representations to get better robustness against specific attacks like PGD. While these methods can be used to gather evidences regarding the robustness of a classifier to first order attacks, i.e., those based on gradient information, they do not exclude the possibility of a stronger attack that can significantly degrade their performance.

There have been recent works studying the problem of efficiently certifying the adversarial robustness of classifiers from both empirical and theoretical perspectives [29, 30, 31, 32, 33, 34, 35, 36, 37]. For the case of certifying robustness to $\ell_\infty$ perturbations, recent works have explored linear programming (LP) and semi-definite programming (SDP) based approaches to produce lower bounds on the certified robustness of neural networks. Such approaches have seen limited success beyond shallow (depth at most 4) networks [13, 15]. These works use convex programming (including SDPs) to directly provide an upper bound on the objective that corresponds to finding an adversarial perturbation for a given classifier $f$ on input $x$. While our multiplicative weights method also approximately solves a convex program (specifically, an SDP), we use the convex program to translate certified $\ell_2$ robustness guarantees to certified $\ell_\infty$ guarantees in a black box manner.

Another line of work has focused on methods for certifying robustness via propagating interval bounds [14]. While these methods have seen recent success in scaling up to large networks, they typically cannot be applied in a black box manner and require access to the structure of the underlying network.

Randomized smoothing proposed in the works of [38, 10, 11, 39] is a simple and effective way to certify robustness of neural networks to $\ell_2$ perturbations. These methods work by creating a smoothed classifier by adding Gaussian noise to inputs. The improved Lipschitzness of the smoothed classifier can then be translated to bounds on certified $\ell_2$ robustness. Other recent works have also proposed developing Lipschitz classifiers in order to get improved robustness [40].

The recent work of [12], that we build upon, shows that by incorporating the smoothed classifier into the training process, one can get state-of-the art results for certified $\ell_2$ robustness. The recent works of [41, 42, 17] extends the work of [12] by developing smoothing based methods for other $\ell_p$ norms and noise distribution other than the Gaussian distribution. The authors in [43] explore smoothing based methods for making a pretrained classifier certifiably robust.

While smoothing based methods have been successful for providing robustness to $\ell_2$ perturbations, they have not demonstrated the same benefits for for the more challenging setting of robustness to $\ell_\infty$ perturbations. The recent works of [16, 17, 18] have provided lower bounds demonstrating that smoothing based methods are unlikely to yield benefits for robustness to $\ell_\infty$ perturbations in the worst case. There have also been recent works exploiting low rank representations to achieve better adversarial robustness [44] empirically against first order attacks such as the PGD method [6]. The goal in our work however, is to use low rank representations to achieve *certified* accuracy guarantees.

*Fast approximate SDP algorithms.* Our certification algorithm is based on the multiplicative weights update (MWU) framework applied to the problem of Quadratic Programming. Klein and Lu [26] gave the first MWU-based algorithm for solving particular semidefinite programs (SDP). They used the framework of [45] for solving convex programs to give a faster algorithm for solving the maxcut SDP within a $1 + \delta$ approximation using $\tilde{O}(n)$ eigenvalue/eigenvector

computations, where the $\tilde{O}$ hides polylogarithmic factors in $n$ and polynomial factors in $1/\delta$. But the analysis of Klein and Lu [26] is specialized for the Max-Cut problem, which is a special case of (11) where $M$ is a graph Laplacian; it does not directly extend to our more general setting (they use diagonal dominance of a Laplacian to get a small bound on the width). Arora, Hazan and Kale [19, 20] introduced a more general framework for solving semidefinite programs. In particular, they showed how when this framework is applied to the problem of Quadratic Programming, it gives a running time bound of $O(\frac{n^{1.5}}{\delta^{2.5}} \cdot \min\{m, n^{1.5}/(\delta\alpha^*)\})$, where the optimal solution value is $\text{SDP}_{val} = \alpha^*\|M\|_1$, and $m$ is the number of non-zeros of matrix $M$. For SDPs of the form (12), the width parameter could be reasonably large and could result in $\tilde{\Omega}(n^{3/2})$ iterations (see Section 6.3 of [20]). To the best of our knowledge this represents the prior best running time for approximately solving the Quadratic Programming SDP (9), and is most related to our work.

Arora et al. [46] also gave primal-dual based algorithms near-linear time algorithms for several combinatorial problems like max-cut and sparsest cut that use semidefinite programming relaxations. Iyengar et al. [47, 48, 49] consider positive covering and packing SDPs like max-cut, and give fast approximate algorithms based on the multiplicative weights method. In particular, [48, 49, 50] and several other works give parallel algorithms, where the iteration count only has a mild polylogarithmic dependence on width for solving these positive SDPs. This uses the matrix multiplicative weights method that involves computing matrix exponentials. Quadratic Programming does not fall into this class of problems in general, unless $M \succeq 0$. A very recent paper of Lee and Padmanabhan [51] gives algorithms that work for problems like quadratic programming with diagonal constraints; however this gives an additive approximation to the objective which does not suffice for our purposes. Moreover, our algorithm is very simple and practical, and each iteration only uses a single computation of the largest eigenvalue. Hence, analyzing this algorithm is interesting in its own right. Finally, interior point methods, cutting plane methods and the Ellipsoid algorithm find solutions to the SDP that have much better dependence on the accuracy $\delta$ e.g., a polynomial in $\log(1/\delta)$ dependence, at the expense of significantly higher dependence on $n$ [52, 53].

# B   Additional experiments for certified $\ell_2$ robustness

We first discuss the setting of the hyperparameters in our experiments. The experiments in Figure 1, Figure 2, and Figure 4 were obtained by training a ResNet-32 architecture [54]. In each case we optimize the objective in (7) with a starting learning rate of 0.1 and decaying the learning rate by a factor of 0.1 every 50 epochs. Each model was trained for 150 epochs. We follow the methodology of [12] and approximate the soft classifier in (5) by drawing 4 noise vectors from the Gaussian distribution $N(0, \sigma^2 I)$ and optimizing the inner maximization in (7) via 10 steps of projected gradient descent. When training the method of [12] we choose the value of $\sigma$ to be 0.12 for $\varepsilon = 0.25, 0.5$. For $\varepsilon = 0.75$ and $\varepsilon = 1.0$, we choose $\sigma$ to be 0.25 and 0.5 respectively. When training our procedure in Algorithm 1 we use a higher noise of $\sigma \cdot \lambda\sqrt{d/r}$, where $\sigma$ is the corresponding noise used for the method of [12], $d = 1024$ and $r = 200$. We vary the parameter $\lambda$ as discussed in Section 2. Empirically, we find that $\lambda = 0.5$ gives the best results.

We first demonstrate that real data such as images have a natural low rank structure. As an illustration, Figure 5 shows the relative reconstruction error for the CIFAR-10 and CIFAR-100 [55] datasets, when each of the three channels is projected onto subspaces of varying dimensions, computed via principal component analysis (PCA). As can be seen, even when projected onto 200 dimensions, the reconstruction error remains small.

The PCA reconstruction error in Figure 5 dictated our choice of $r = 200$. This choice made apriori to the training stage works well across different settings. Changing the rank $r$ by small amounts had negligible effect on the certified accuracy (we tried values of $r$ close to 200). The smaller we can keep $r$, the more robustness we can achieve. Additionally, the performance of our algorithm is smooth in $\lambda$ (see Figure 1). Furthermore, in all our experiments $\lambda = 0.5$ worked very well pointing to the fact that there are generic settings of $\lambda$ that one can often use.

R channel of CIFAR-10   G channel of CIFAR-10   B channel of CIFAR-10
R channel of CIFAR-100   G channel of CIFAR-100   B channel of CIFAR-100

Figure 5

Next we provide in Figure 6 a more fine grained comparison between our procedure in Algorithm 1 and the method of [12] by separately comparing the performance of the two methods for different values of $\varepsilon$ on the CIFAR-10 and CIFAR-100 datasets.

## C   Simple Theoretical Propositions and Proofs

The following proposition shows the equivalence between the smoothed classifiers in (4) and (5).

**Proposition C.1.** *Given a base classifier $f : \mathbb{R}^n \to \mathcal{Y}$ and a projection matrix $\Pi$, on any input $x$, the smoothed classifier $g_\Pi(x)$ as defined in (4) is equivalent to the classifier given by*

$$\tilde{g}_\Pi(x) = \arg \max_{y \in \mathcal{Y}} \mathbb{P}(f\big(\Pi(x + \delta)\big) = y).$$

*Here $\delta$ is a standard Gaussian of variance $\sigma^2$ in every direction.*

*Proof.* Let $\Pi$ be a projection matrix with the corresponding subspace denoted by $\mathcal{S}$. Let $\delta$ be a random variable distributed as $N(0, \sigma^2 I)$ and $\delta_\Pi$ be a standard normal random variable with variance $\sigma^2$ within $\mathcal{S}$ and variance 0 outside. From the property of Gaussian distributions we have that projections of a spherical Gaussian random variable with variance $\sigma^2$ in each direction are themselves Gaussian random variables with variance $\sigma^2$ in each direction within the subspace. Hence, for a fixed input $x$, the random variables $\Pi x + \delta_\Pi$ and $\Pi x + \Pi \delta$ are identically distributed. From this we conclude that the classifier

$$g_\Pi(x) = \arg \max_{y \in \mathcal{Y}} \mathbb{P}(f(\Pi x + \delta_\Pi) = y).$$

is identical to the classifier $\tilde{g}_\Pi(x)$.   $\square$

The following proposition shows how to obtain an $\ell_\infty$ robustness guarantee from an $\ell_2$ robustness guarantee.

**Proposition C.2.** *Consider any classifier $f : \mathbb{R}^n \to \mathcal{Y}$, and let the classifier $g(x) := f(\Pi x)$, where $\Pi \in \mathbb{R}^{n \times n}$ is any projection matrix. Then if we denote by $r_2(x), r_\infty(x)$ the radius of robustness measured in $\ell_2$ and $\ell_\infty$ norm respectively of $g$ around a point $x \in \mathbb{R}^n$. Then we have*

$$r_\infty(x) \geq \frac{r_2(x)}{\|\Pi\|_{\infty \to 2}}, \ where \ \|\Pi\|_{\infty \to 2} = \max_{x : \|x\|_\infty \leq 1} \|\Pi x\|_2.$$

*Hence for any $\varepsilon \geq 0$, we have $acc_{\varepsilon'}^{(\ell_\infty)}(g) \geq acc_\varepsilon^{(\ell_2)}(g)$, for all $0 \leq \varepsilon' \leq \varepsilon / \|P\|_{\infty \to 2}$.*

Figure 6: A comparison of certified radius guarantees obtained via Algorithm 1 as compared to the approach of [12]. The x-axis is the radius, and the y-axis represents the certified accuracy.

*Proof.* Consider any perturbation $x' = x + z$ where $\|z\|_\infty \leq \varepsilon'$ such that the predictions given by $g(x') = f(\Pi x')$ and $g(x) = f(\Pi x)$ differs. Consider another perturbation $z'$ such that $z' = \Pi z$. Notice that $g(x + z') = f(\Pi(x + \Pi z)) = g(x')$, since $\Pi^2 = \Pi$ for a projection matrix. Hence, the predictions of the network differ on $x$ and $x + z'$. Moreover $\|z'\|_2 \leq \|\Pi\|_{\infty \to 2}\varepsilon'$. Hence $g$ is not robust at $x$ up to an $\ell_2$ radius of $\varepsilon = \varepsilon'\|\Pi\|_{\infty \to 2}$, as required. $\square$

The following simple proposition shows how to obtain certified robustness guarantees in the representation $\mathcal{U}$ e.g., DCT basis by performing appropriate training in another representation $\mathcal{X}$ e.g., the co-ordinate or pixel basis. Let $O \in \mathbb{R}^{n \times n}$ represent an orthogonal matrix that represents the DCT transformation i.e., for an input $x \in \mathcal{X} \subseteq \mathbb{R}^n$ let $\psi(x) = Ox$ represents its DCT representation (hence $\mathcal{U} = \psi(\mathcal{X})$).

**Proposition C.3.** *Suppose $\mathcal{X}, \mathcal{U} = \psi(\mathcal{X})$ be defined as above. For any classifier $g : \mathcal{U} \to \mathcal{Y}$, consider the classifier $f : \mathcal{X} \to \mathcal{Y}$ obtained as $f = g(Ox)$ where $O$ is an orthogonal matrix. For a point $x \in \mathcal{X}$ if we denote by $r_f^{(\ell_2)}(x)$ the radius of robustness of $f$ at $x$ measured in $\ell_2$ norm, then we have that $r_g^{(\ell_2)}(Ox) = r_f^{(\ell_2)}(x)$. Moreover suppose the classifier $g(u) = g(\Pi u) \; \forall u \in \mathcal{U}$, for some projection matrix $\Pi$, then the robust accuracy of $g$ to $\ell_\infty$ perturbations in the representation $\mathcal{U}$ satisfies $\text{acc}_{\varepsilon'}^{(\ell_\infty)}(g) \geq \text{acc}_\varepsilon^{(\ell_2)}(f)$ for any $\varepsilon > 0$ and $\varepsilon' \leq \varepsilon/\|\Pi\|_{\infty \to 2}$.*

*Proof.* We will prove by contradiction. Let $u, u' \in \psi(\mathcal{X})$ and let $u' = u + \delta_u$ be an adversarial perturbation of $u$ with $\|\delta_u\|_2 = r$. Let $x = O^{-1}u$, and let $x' = O^{-1}u'$; note that $x, x'$ exist since $u, u' \in \psi(\mathcal{X})$. Moreover $\|x - x'\|_2 = \|u - u'\|_2$ since $O$ is a rotation (orthogonal matrix). Hence $x'$ is an adversarial example for $f$ at $x$, at a $\ell_2$ distance of $\delta_u$. This shows that $r_g^{(\ell_2)}(Ox) \geq r_f^{(\ell_2)}(x)$. A similar argument also shows that $r_g^{(\ell_2)}(Ox) \leq r_f^{(\ell_2)}(x)$. The last part of the proposition follows by applying Proposition C.2. $\square$

The following simple fact relates the $\infty \to 2$ operator norm and the $\ell_1$ sparsity of a projection matrix. This justifies the use of $\ell_1$ sparsity as an approximate relaxation or proxy for $\|\Pi\|_{\infty \to 2}$.

**Fact C.4.** *For any (orthogonal) projection matrix $\Pi \in \mathbb{R}^{n \times n}$ of rank $r$, we have*

$$\frac{\|\Pi\|_1}{r} \leq \|\Pi\|_{\infty \to 2}^2 = \|\Pi\|_{\infty \to 1} \leq \|\Pi\|_1,$$

*where $\|\Pi\|_1$ refers to the entry-wise $\ell_1$ norm of $\Pi$.*

*Proof.* First we show the upper bound

$$\|\Pi\|_{\infty \to 1} = \max_{y, z \in \mathbb{R}^n : \|y\|_\infty \leq 1, \|z\|_\infty \leq 1} \langle \Pi, yz^\top \rangle \leq \sum_{i,j} |\Pi(i, j)| = \|\Pi\|_1.$$

We now show the lower bound. Let $\Pi = \sum_{i=1}^r v_i v_i^\top$, where $\{v_i : i \in [r]\}$ represents an orthonormal basis for the subspace given by $\Pi$. We have from the monotonicity of $\infty \to 1$ operator norm shown in [21] that

$$\|\Pi\|_{\infty \to 1} = \Big\|\sum_{i=1}^r v_i v_i^\top\Big\|_{\infty \to 1} \geq \max_{i \in [r]} \|v_i v_i^\top\|_{\infty \to 1} = \max_{i \in [r]} \|v_i v_i^\top\|_1$$

$$\geq \frac{1}{r} \sum_{i \in [r]} \|v_i v_i^\top\|_1 \geq \frac{1}{r} \Big\|\sum_{i \in [r]} v_i v_i^\top\Big\|_1 = \frac{\|\Pi\|_1}{r},$$

where the last inequality uses the triangle inequality for matrix operator norms.

$\square$

# D   Translating certified adversarial robustness from $\ell_2$ to $\ell_\infty$ perturbations

## D.1   Certifying the $\infty \to 2$ operator norm

We now describe our efficient algorithmic procedure that gives an upper bound on the $\infty \to 2$ norm of the given matrix. Moreover, as we will see in Theorem D.2, our algorithmic procedure comes with provable guarantees: it is guaranteed to output a value that is only a small constant factor off from the global optimum i.e., true value of the $\infty \to 2$ norm.

For any matrix $A$, we first note that $\|A\|_{\infty \to 2}^2 = \|A^\top A\|_{\infty \to 1}$. Moreover, by the variational characterization of operator norms, we have for $M = AA^\top \succeq 0$

$$\|A\|_{\infty \to 2}^2 = \max_{x:\|x\|_\infty \leq 1} \max_{y:\|y\|_\infty \leq 1} x^\top My = \max_{x:\|x\|_\infty \leq 1} x^\top Mx. \tag{11}$$

As stated in Section 3, our algorithm will apply to the more general problem (9) where $M_{ii} \geq 0$ for all $i \in [n]$.

$$\text{Given a symmetric matrix } M \text{ with } \forall i \in [n]: M_{ii} \geq 0, \qquad \max_{x:\|x\|_\infty \leq 1} x^\top Mx.$$

Note that this is certainly satisfied by all $M \succeq 0$. We consider the following standard SDP relaxation (12) for the problem, where $M \succeq 0$. The primal variables are represented by the positive semi-definite (PSD) matrix $X \in \mathbb{R}^{n \times n}$. The SDP dual of this relaxation, given in (14), has variables $y_1, \ldots, y_n \geq 0$ corresponding to the $n$ constraints (13). In what follows, for matrices $M, X$, $\langle M, X \rangle := \mathrm{tr}(M^\top X)$ represents the trace inner product.

| **Primal SDP:** $\max_X \langle M, X \rangle$ (12) | **Dual SDP:** $\min_y \sum_{i \in [n]} y_i$ (14) |
|---|---|
| s.t. $X_{ii} \leq 1, \quad \forall i \in [n]$ (13) | s.t. $diag(y) \succeq M$ (15) |
| $X \succeq 0.$ | $y \geq 0.$ |

Let us denote by $\mathrm{SDP}_{val}$ the value of the optimal solution to the primal SDP (12). Recall that our algorithm goal is to output a value that is guaranteed to be a valid upper bound for $\|A\|_{\infty \to 2}^2 = \|M\|_{\infty \to 1}$.[3] The above primal SDP (12) is a valid relaxation, and it is also tight up to a factor of $\pi/2$ i.e., $(2/\pi)\mathrm{SDP}_{val} \leq \|M\|_{\infty \to 1} \leq \mathrm{SDP}_{val}$. Moreover by weak duality, any feasible solution to the dual SDP (14) gives a valid upper bound for the primal SDP value, and hence $\|M\|_{\infty \to 1}$. However, the above SDP is computationally intensive to solve using off-the-shelf SDP solvers (even for CIFAR-10 images, $X$ is $1024 \times 1024$). Instead we design an algorithm based on the *multiplicative weight update* (MWU) framework for solving SDPs [26, 19].

Our algorithm is described in Algorithm 2 of Section 3. Our algorithm differs from the standard MWU approach (and analysis) [20] that treats the constraints (13) and the objective (12) similarly. Firstly, by treating the objective differently from the constraints, we ensure that the width of the convex program is smaller; in particular, the "width" parameter does not depend on the value of the objective function anymore. This allows us to get a better convergence guarantee. Secondly, we always maintain a dual feasible solution that gives a valid upper bound on the objective value (this also allows us to do an early stop if appropriate). We remark that for *every* choice of $\alpha$ (and the update rule), $\mathrm{val}_{curr} = n\lambda$ gives a valid upper bound (due to dual feasibility). This is encapsulated in the following simple proposition where $\mathrm{SDP}_{val}$ refers to the optimal solution value of the (12).

**Proposition D.1.** *For any $\alpha \in \mathbb{R}_{\geq 0}^n$ with $\sum_{i=1}^n \alpha(i) = n$, if $\lambda$ is the maximum eigenvalue*

$$\lambda = \lambda_{\max}\Big( \big( diag(\alpha)^{-1/2} M\, diag(\alpha)^{-1/2} \big) \Big), \text{ then we have } \mathrm{SDP}_{val} \leq n\lambda,$$

*and $y = \lambda\alpha$ is feasible for the dual SDP (14) and attains an objective value of $n\lambda$.*

*Proof.* Consider $y = \lambda\alpha$. Firstly, the dual objective value at $y$ is $\sum_{i=1}^{n} y_i = \lambda \sum_i \alpha(i) = n\lambda$, as required. Moreover $y \geq 0$, since the $\alpha \geq 0$. Finally, (15) is satisfied since by definition of $\lambda$,

$$\operatorname{diag}(\alpha)^{-1/2} M \operatorname{diag}(\alpha)^{-1/2} \preceq \lambda \cdot I \implies M \preceq \lambda \cdot \operatorname{diag}(\alpha) = \operatorname{diag}(y),$$

by pre-multiplying and post-multiplying by $\operatorname{diag}(\alpha)^{1/2}$. Finally, from weak duality $\mathrm{SDP}_{val} \leq n\lambda$. $\square$

**Analysis of the algorithm**   We show the following guarantee for our algorithm.

**Theorem D.2.** *For any $\delta > 0$, any symmetric matrix $M$ with $M_{ii} \geq 0 \ \forall i \in [n]$ , Algorithm 2 on input $M$, with parameters $\delta$ and $\rho = O(n/\delta)$ after $T = O(n \log n/\delta^3)$ iterations finds a solution $\widehat{X} \in \mathbb{R}^{n \times n}$ and $\widehat{y} \in \mathbb{R}^n$ such that*

*(a) $\widehat{y}$ is feasible for the dual SDP (14) such that $\langle M, \widehat{X}\rangle = \sum_{i=1}^{n} y_i$.*
*(b) $\widehat{X} \succeq 0$, $\widehat{X}_{ii} \leq 1 + \delta$ for all $i \in [n]$, and $\langle M, \widehat{X}\rangle \geq SDP_{val}$, the primal SDP value.*

The above Theorem D.2 and Proposition D.1 together imply Theorem 3.1. We remark that the $\tilde{O}(n \log n) \cdot T_{eig}$ running time guarantee almost matches the guarantees for Klein and Lu for Max-Cut SDP [26] (up to a $O(1/\delta)$ factor), but our guarantees apply to the SDP for the more general Quadratic Programming problem. Note that the maximum eigenvalue can be computed within $\delta$ accuracy in $O(m/\delta)$ time where $m$ is the number of non-zeros of $M$, (see e.g., [26] for a proof and use in MWU framework). To the best of our knowledge, the best known algorithm prior to our work for solving the SDP to this general Quadratic Programming problem (or even problem (11)) is by Arora, Hazan and Kale [19, 20] who give a $O(\frac{n^{1.5}}{\delta^{2.5}} \cdot \min\{m, n^{1.5}/(\delta\alpha^*)\})$, where the optimal solution value is $\mathrm{SDP}_{val} = \alpha^* \|M\|_1$. Compared to our algorithm's running time of $\tilde{O}(nm \log n)$, even when $\alpha^* = \Omega(1)$ the previous best requires a running time of $\tilde{O}(n^{1.5} \min\{m, n^{1.5}\})$; but it can even be the case that $\alpha^* = O(1/n)$ [56]. Finally recall that upper bound given by the SDP is only off by a factor of $\pi/2$ for PSD matrices, that is the best possible assuming $P \neq NP$ (for general matrices the approximation factor could be $O(\log n)$) [24, 56, 25, 57].

The analysis closely mirrors the analysis of the Multiplicative Weights algorithm for solving SDPs due to Klein and Lu [26], and Arora, Hazan and Kale [19, 20]. The main parameter that affects the running time of the multiplicative weights update method is the width parameter $\rho'$ of the SDP constraints. The analysis of Klein and Lu [26] is specialized for the Max-Cut problem (which is a special case of (11) where $M$ is a graph Laplacian), where they achieve a bound of $O(n \log n/\delta^2)$ eigenvalue computations. However their analysis does not directly extend to our more general setting (they crucially use the fact that the optimum solution value is at least $\Omega(\mathrm{tr}(M))$ to get a small bound on the width). In the more general framework of [19], the optimization problem is first converted into a feasibility problem (hence the objective also becomes a constraint). For SDPs of the form (12), the width parameter could be reasonably large and could result in $\Omega(n^{3/2})$ iterations (see Section 6.3 of [20]). In our analysis, we ensure that the width is small by treating the objective separately from the constraints. A crucial step of our algorithm is introducing an amount $\delta$ to each of the weights which solving the eigenvalue maximization in each step. This correction term of $\delta$ is important in getting an upper bound on the violation of each constraint, and hence the width of the program. Finally, we present a slightly different analysis using the SDP dual, that has the additional benefit of providing a certificate of optimality.

**Proof of Theorem D.2.**   Let at step $t$ of the iteration, $\alpha^{(t)}$ be the weight vector, $X^{(t)}$ be the candidate SDP solution and $y^{(t)}$ be the candidate dual solution maintained by the Algorithm 2 in step 6. Also $\tilde{\alpha}^{(t)} := (1 - \delta)\alpha^{(t)} + \delta\mathbf{1}$, where $\mathbf{1} = (1, 1, \ldots, 1)$. It is easy to see that

$$\forall t \leq T, \quad X^{(t)} = \frac{1}{t}\sum_{\ell=1}^{t} v^{(\ell)}(v^{(\ell)})^\top, \quad \text{and} \quad y^{(t)} = \frac{1}{t}\sum_{\ell=1}^{t}\lambda^{(\ell)}\tilde{\alpha}^{(\ell)}.$$

We first establish the following lemma, which immediately implies part (a) of the theorem statement.

**Lemma D.3.** *For all $t \le T$, $y^{(t)}$ is feasible for the dual SDP (14), and $\langle M, X^{(t)} \rangle = \sum_i y_i^{(t)}$.*

*Proof.* We first check feasibility of the dual solution. First note that in every iteration $t \in [T]$, $\tilde{\alpha}^{(t)} \ge 0$ and $\lambda^{(t)} \ge 0$. The latter is because $\operatorname{tr}(\operatorname{diag}(\tilde{\alpha}^{(t)})^{-1/2} M \operatorname{diag}(\tilde{\alpha}^{(t)})^{-1/2}) \ge 0$ as all the diagonal entries of $M$ are non-negative, and hence $\lambda_{\max}(\operatorname{diag}(\tilde{\alpha}^{(t)})^{-1/2} M \operatorname{diag}(\tilde{\alpha}^{(t)})^{-1/2}) \ge 0$. Hence $y^{(t)} \ge 0$. Moreover, by definition of $\lambda^{(t)}$

$$M \preceq \lambda^{(\ell)} \cdot \operatorname{diag}(\tilde{\alpha}^{(\ell)}) \ \ \forall \ell \le T, \ \ (\text{ since } \operatorname{diag}(\tilde{\alpha}^{(t)})^{-1/2} M \operatorname{diag}(\tilde{\alpha}^{(t)})^{-1/2} \preceq \lambda^{(t)} I),$$

$$M \preceq \frac{1}{t} \sum_{\ell=1}^{t} \lambda^{(\ell)} \cdot \operatorname{diag}(\tilde{\alpha}^{(\ell)}) = \frac{1}{t} \sum_{\ell=1}^{t} \operatorname{diag}(y^{(\ell)}).$$

Finally, we establish $\langle M, X^{(t)} \rangle = \sum_i y_i^{(t)}$. Note that by definition $v^{(\ell)} = \sqrt{n} \operatorname{diag}(\tilde{\alpha}^{(\ell)})^{-1/2} u^{(\ell)}$, where $u^{(\ell)}$ is the top eigenvector of $(\operatorname{diag}(\tilde{\alpha})^{-1/2} M \operatorname{diag}(\tilde{\alpha})^{-1/2})$, and $\lambda^{(\ell)}$ is its eigenvalue. Hence

$$\langle M, X^{(t)} \rangle = \frac{1}{t} \sum_{\ell=1}^{t} \langle M, v^{(\ell)}(v^{(\ell)})^\top \rangle = \frac{1}{t} \sum_{\ell=1}^{t} (v^{(\ell)})^\top M v^{(\ell)}$$

$$= n \frac{1}{t} \sum_{\ell=1}^{t} \left( (u^{(\ell)})^\top \operatorname{diag}(\tilde{\alpha}^{(\ell)})^{-1/2} M \operatorname{diag}(\tilde{\alpha}^{(\ell)})^{-1/2} u^{(\ell)} \right) = \frac{1}{t} \sum_{\ell=1}^{t} \lambda^{(\ell)} \cdot n$$

On the other hand, $\displaystyle\sum_{i=1}^{n} y_i^{(t)} = \sum_{i=1}^{n} \frac{1}{t} \sum_{\ell=1}^{t} \lambda^{(\ell)} \tilde{\alpha}^{(\ell)}(i) = \frac{1}{t} \sum_{\ell=1}^{t} \lambda^{(\ell)} \cdot \sum_{i=1}^{n} \tilde{\alpha}^{(\ell)}(i) = \frac{1}{t} \sum_{\ell=1}^{t} \lambda^{(\ell)} \cdot n.$

This proves the lemma. □

We now complete the proof of Theorem D.2.

*Proof.* Recall that $\widehat{X} = X^{(T)}, \widehat{y} = y^{(T)}$. Consider the SDP solution $X' = \frac{1}{1+\delta} \widehat{X}$. From Lemma D.3, we see that $y^{(T)}$ is feasible and

$$\langle M, X' \rangle = \frac{1}{1+\delta} \langle M, \widehat{X} \rangle = \frac{1}{1+\delta} \sum_{i=1}^{n} \widehat{y}(i) \ge \frac{1}{1+\delta} \cdot \text{SDP}_{val},$$

where the last inequality follows from weak duality of the SDP. Hence it suffices to show that $X'$ is feasible i.e., $\widehat{X}_{ii} \le 1 + \delta$ for all $i \in [n]$.

We prove $\max_{i \in [n]} \widehat{X}_{ii} \le 1 + \delta$ by following the same multiplicative weight analysis in [20] (see Section 2, Theorem 5), but only restricted to the $n$ constraints of the form $X_{ii} \le 1$.

There is one expert for each $i \in [n]$ corresponding to the constraint $X_{ii} \le 1$. At each iteration $t$, we consider a probability distribution $p^{(t)} = \frac{1}{n} \alpha^{(t)}$. The loss of expert $i$ at time $t$ is given by $m^{(t)}(i) = \frac{1}{\rho}(1 - (v_i^{(t)})^2)$ for $\rho = n/\delta$ . We note that in each iteration $\ell$,

$$\sum_i (\alpha_i^{(t)} + \delta)(v_i^{(t)})^2 \le n \implies \forall i \in [n], \ (v_i^{(t)})^2 \le \frac{n}{\delta} \le \rho$$

Hence $\dfrac{1}{\rho} \ge m_i^{(t)} = \dfrac{1}{\rho}\left(1 - (v_i^{(t)})^2\right) \ge -\dfrac{(n/\delta - 1)}{\rho} \ge -1.$

From the guarantees of the multiplicative weight update method (see Section 2, Theorem 2 of [20]), we have for each $i \in [n]$

$$\sum_{t=1}^{T}\sum_{i=1}^{n} m_i^{(t)} p_i^{(t)} \leq \sum_{t=1}^{T} m_i^{(t)} + \delta \sum_{t=1}^{T} |m_i^{(t)}| + \frac{\ln n}{\delta}$$

$$= (1-\delta)\sum_{t=1}^{T} m_i^{(t)} + 2\delta \sum_{\substack{t \in [T] \\ m_i^{(t)} > 0}} |m_i^{(t)}| + \frac{\ln n}{\delta}$$

$$\leq \frac{(1-\delta)}{\rho}\sum_{t=1}^{T} \left(1 - (v_i^{(t)})^2\right) + \frac{2\delta T}{\rho} + \frac{\ln n}{\delta}$$

$$\frac{\rho}{T}\sum_{t=1}^{T}\sum_{i=1}^{n} m_i^{(t)} p_i^{(t)} \leq (1-\delta) \cdot (1 - \widehat{X}_{ii}) + 2\delta + \frac{\rho \ln n}{T\delta}. \tag{16}$$

On the other hand, using the fact that $\sum_i \alpha_i^{(t)} = n$, and $\sum_i \tilde{\alpha}_i^{(t)}(v_i^{(t)})^2 = n$ we have

$$\frac{\rho}{T}\sum_{t=1}^{T}\sum_{i=1}^{n} m_i^{(t)} p_i^{(t)} = \frac{1}{T}\sum_{t=1}^{T}\frac{1}{n}\sum_{i=1}^{n} \left(1 - (v_i^{(t)})^2\right)\alpha_i^{(t)}$$

$$= \frac{1}{T}\sum_{t=1}^{T}\frac{1}{n}\sum_{i=1}^{n} \alpha_i^{(t)} - \frac{1}{T}\sum_{t=1}^{T}\frac{1}{n}\sum_{i=1}^{n} \alpha_i^{(t)}(v_i^{(t)})^2 = -\frac{1}{T}\sum_{t=1}^{T}\frac{1}{n}\sum_{i=1}^{n} \left(\frac{\tilde{\alpha}_i^{(t)}}{1-\delta} + \frac{\delta}{1-\delta}\right)(v_i^{(t)})^2 + 1$$

$$= -\frac{1}{T}\sum_{t=1}^{T}\frac{1}{(1-\delta)n}\sum_{i=1}^{n} \tilde{\alpha}_i^{(t)}(v_i^{(t)})^2 + \frac{1}{T}\sum_{t=1}^{T}\frac{1}{(1-\delta)n}\sum_{i=1}^{n} \delta(v_i^{(t)})^2) + 1$$

$$= -\frac{1}{1-\delta} + 1 + \frac{1}{T}\sum_{t=1}^{T}\frac{1}{(1-\delta)n}\sum_{i=1}^{n} \delta(v_i^{(t)})^2$$

$$\geq -\frac{\delta}{1-\delta} \tag{17}$$

Combining (16) and (17), we get that for $\delta \in (0, 1/2)$, and $T = \rho \log n/((1-\delta)\delta^2) = O(n \log n/\delta^3)$

$$\forall i \in [n], \quad (1 - \widehat{X}_{ii}) \geq -\frac{\delta}{(1-\delta)^2} - \frac{2\delta}{1-\delta} - \frac{\rho \ln n}{T\delta(1-\delta)}$$

$$\widehat{X}_{ii} \leq 1 + \delta(1 + 6\delta) + 2\delta(1 + 2\delta) + \delta \leq 1 + 9\delta.$$

This completes the proof of part (b). It is also straightforward to see that in above analysis, a $(1 + \delta)$ approximate eigenvalue method can also be used to get a similar guarantee with an extra $(1 + \delta)$ factor loss in the objective value. $\qquad\square$

We remark that the larger dependence of $\rho = O(n/\delta)$ is needed to ensure that in each iteration $(v_i^{(t)})^2/\rho \leq 1$. If this condition is satisfied for $\rho \ll n/\delta$, then the iteration bound is $O(\rho \log n)/\delta^2$. This also justifies the use of a more aggressive i.e., smaller choice of $\rho$ in practice.

## D.2   Finding Robust Low-rank Representations

We now show how to find a good low-rank robust projections when it exists in the given representation. Given a dataset $A$, our goal is to find a (low-rank) projection $\Pi$ that gets small reconstruction error $\|A - \Pi A\|_F^2$, while ensuring that $\|\Pi\|_{\infty \to 2}$ is small. Awasthi et al. [21] formulate this as an optimization problem that is NP-hard, but show polynomial time algorithms based on the Ellipsoid algorithm that gives constant factor approximations. However, the algorithm is impractical in practice because of the Ellipsoid algorithm, and the separation oracle used by it (that in turn involves solving an SDP). We instead use the

Figure 7: The images on the left correspond to the original images and the images on the right correspond to imperceptible adversarial examples within an $\ell_\infty$ radius of $\varepsilon \leq 0.09$.

connection to sparsity described in Section 3 to devise a much faster heuristic based on sparse PCA to find a good projection $\Pi$ (see (C.4) in the appendix for a formal justification). We just use an off-the-shelf heuristic for sparse PCA (the scikit-learn sparse PCA implementation [58] based on [22]), along with our certification procedure Algorithm 2).

---

**Algorithm 3** Find a Robust Projection

---

1: **function** ROBUSTPROJECTION(data $A \in R^{m \times n}$, rank $k$, reconstruction error $\delta$)
2:      Set $M := (A^\top A)/\text{tr}(A^\top A)$. Initialize $\widehat{\Pi} \leftarrow \emptyset, \widehat{\kappa} = \infty$.
3:      **for** different values of $r \leq k$ **do**
4:          Find $r$-PCA of the $M$ to get a rank $r$ projection $\Pi_1$. $M' \leftarrow M - \Pi_1 M \Pi_1$
5:          Run sparse PCA on $M'$ to find a rank $(k - r)$ projection $\Pi_2$. Set $\Pi = \Pi_1 + \Pi_2$.
6:          Run CERTIFYSDP$(\Pi, \delta = 1/4, \rho)$ to get an upper bound $\kappa$.
7:          **if** $\kappa < \widehat{\kappa}$ and if $\langle M, I - \Pi \rangle \leq \delta$ **then**
8:              $\widehat{\Pi} \leftarrow \Pi, \widehat{\kappa} = \kappa$.
9:      Output $\widehat{\Pi}, \widehat{\kappa}$.

---

# E   Imperceptibility in the DCT basis and Training Certified $\ell_\infty$ Robust Networks

**Adversarial Examples for CIFAR-10 images.** We take a ResNet-32 network that has been trained on the CIFAR-10 datasets via the PGD based method of [6] for robustness to $\ell_\infty$ perturbations. We then generate imperceptible adversarial examples for the test images via projected gradient ascent and an $\ell_\infty$ perturbation radius of $\varepsilon = 0.09$. See Figure 7 for a few of the original images and the corresponding adversarial perturbations.

# F   Experimental Evaluation of MWU-based SDP Certification Algorithm 2

In this section we empirically evaluate the effectiveness of our multiplicative weights based procedure from Algorithm 2 for solving the general Quadratic Programming (QP) problem as defined in (9). Recall that this corresponds to the following optimization problem:

$$\text{Given a symmetric matrix } M \text{ with } \forall i \in [n]: M_{ii} \geq 0, \quad \max_{x: \|x\|_\infty \leq 1} x^\top M x. \qquad (18)$$

Recall that in our Algorithm 2 every iterations involves a single maximum eigenvalue computation, for which we use an off the shelf subroutine from Python's scipy package. In our implementation of Algorithm 2 we add an early stopping condition if the dual value does

Figure 8: Comparison of the running time of Algorithm 2(left plot) with the MOSEK solver for PSD matrices of varying sizes ($n$ from 500 to 4500). The middle plot shows the SDP values output by the two procedures. The right plot shows the relative error in the SDP value output by our procedure as compared to the value output by the MOSEK solver.

| n | Running time (in s) of our Algorithm 2 | Running time (in s) of MOSEK |
|---|---|---|
| 500 | $1.384 \pm 0.324$ | $8.444 \pm 0.389$ |
| 1000 | $5.631 \pm 1.190$ | $66.578 \pm 5.432$ |
| 2000 | $35.747 \pm 8.960$ | $620.615 \pm 161.96$ |
| 3000 | $143.07 \pm 17.29$ | $1387.69 \pm 398.26$ |
| 4000 | $351.81 \pm 142.97$ | $3453.95 \pm 818.50$ |
| 4500 | $518.56 \pm 27.81$ | $5579.93 \pm 523.53$ |

Table 2: The average time (in seconds) taken by Algorithm 2 and the MOSEK solver on random PSD matrices for a few different sizes ($n$). The mean and standard deviation are computed over 5 independent runs for each value of $n$. See also Figure 8 for the plot based on more values of $n$ taken in increments of 250.

not improve noticeably in successive rounds. Recall that Proposition D.1 proves that this also gives a valid upper bound on the primal SDP value.

We consider two scenarios, one where $M$ is a positive semi-definite matrix (PSD) and the other when $M$ is a symmetric matrix with non-negative diagonal entries. In each case we compare the running time of our algorithm and the dual SDP value that it outputs with the corresponding values obtained by using a state of the art SDP solver on the same instance. As a comparison for our experiments we choose the SDP solver from the commercial optimization software MOSEK [59]. For consistency, the comparison experiments were run on a single core of a Microsoft Surface Pro 3 tablet/computer with Intel Core i7-8650U CPU 1.90 GHz with 16GB RAM.

For the case when $M$ is a PSD matrix, we generate a random $n \times n$ matrix $A$ that contains entries drawn from a standard normal distribution and set $M = AA^\top$, and renormalize the matrix so that the trace is 1. We vary $n$ from 500 to 4500 in increments of 250 (we used 5 random trials for each value of $n$). Figure 8 shows the comparison of our algorithm with the SDP solver from MOSEK for different values of $n$, averaged over the trials along with error bars. Notice that the running time of our algorithm is an order of magnitude faster than the MOSEK solver particularly for larger values of $n$, and furthermore the SDP values output are within 0.5% of the values output by exactly optimizing the SDP via the MOSEK solver. We also include a table of the average run times for a few different values of $n$ in Table 2. We remark that other non-commercial SDP solvers like CVXPY [60] timed out even for for $n > 800$.

A similar trend holds in Figure 9 where we consider randomly generated instances of the more general QP problem (18). Here $M$ is chosen to be a random symmetric $n \times n$ matrix with entries drawn from the standard normal distribution and replace the diagonal entries of $M$ with their absolute values (so that the diagonals are non-negative).

Furthermore, unlike MOSEK, our procedure can scale to much larger values of $n$. As an example, Table 3 shows the running time needed for our procedure to perform 200 iterations on PSD matrices $M$ of sizes ranging from 5000 to 2000. This demonstrates the scalability of our method for larger datasets with higher dimensions. For scalability purposes, the experiments below were run on a machine with access to a single GPU. Hence, the runtimes reported in Table 3 are not directly comparable to those in Table 2.

Figure 9: Comparison of the running time of Algorithm 2(left plot) with the MOSEK solver for general symmetric matrices with non-negative diagonal entries of varying sizes ($n$). The middle plot shows the SDP values output by the two procedures. The right plot shows the relative error in the SDP value output by our procedure as compared to the value output by the MOSEK solver.

| n | Running time (in s) of our Algorithm 2 |
|---|---|
| 5000 | $104.92 \pm 4.04$ |
| 6000 | $156.29 \pm 4.99$ |
| 7000 | $213.80 \pm 7.86$ |
| 8000 | $272.83 \pm 17.29$ |
| 9000 | $346 \pm 8.99$ |
| 10000 | $378.70 \pm 17.4$ |
| 15000 | $1092.14 \pm 103.79$ |
| 20000 | $1801.08 \pm 63.95$ |

Table 3: The time (in seconds) taken by Algorithm 2 to perform 200 iterations on random PSD matrices of varying sizes ($n$). The mean and standard deviation are computed over 5 independent runs for each value of $n$.

We remark that our algorithm is specific to the Quadratic Programming SDP (which is a large class of problems in itself), while MOSEK SDP solver is a general purpose commercial SDP solver based on interior-point methods that is more accurate. Our MWU-based algorithm (Algorithm 2) for the Quadratic Programming SDP (12) gives much faster algorithms for approximately solving the SDP (along with dual certificate of the upper bounds), while not compromising much on approximation loss in the value. Hence the experiments are consistent with the running time improvements suggested by the theoretical results in Section 3.

## G   Robust Projections for Audio Data in DCT basis

We now perform an empirical evaluation on audio data to demonstrate the existence of good low-dimensional robust representations in the DCT basis. We consider the Mozilla CommonVoice speech-to-text audio dataset (english en_1488h_2019-12-10)[4] that was also considered by the work of Carlini et al. [3] on audio adversarial examples. We use a standard approach (that is also employed by [3]) to convert each audio file that may be of variable duration into a representation in high dimensional space. Each element $x_i$ is a signed 16-bit

value. First each audio file is converted into a sequence of overlapping frames corresponding to time windows; and each frame is represented as an $n$-dimensional vector of 16 bit values where $n$ is given by the product of the window size and the sampling rate (for the Commonvoice dataset $n = 1200$). Hence each audio file corresponds to a sequence of frames, each of which is represented by an $n$ dimensional vector.

| Trial number | Number of frames N | $\infty \to 2$ norm | Projection error % |
|:---:|:---:|:---:|:---:|
| 1 | 421469 | 21.285 | 4.974 |
| 2 | 480773 | 22.899 | 5.056 |
| 3 | 447353 | 21.962 | 5.578 |
| 4 | 461034 | 24.439 | 5.247 |
| 5 | 438954 | 23.840 | 5.392 |
| 6 | 452581 | 22.902 | 5.025 |

Table 4: The table shows SDP upper bounds on $\infty \to 2$ norm for projection matrices of rank $r = 200$ obtained by Algorithm 3 for 6 trials each with random 1000 audio samples from the Mozilla Common Voice dataset. Each random sample of 1000 audio samples corresponds to roughly $N \approx 10000$ frames each of $n = 1200$ dimension in the DCT basis.

For our experiments, in each random trial we consider 1000 randomly chosen audio files, and consider the data matrix (with $n$-dimensional columns) consisting of all the frames corresponding to these audio files (and we consider 6 such random trials). We first use the sparse PCA based heuristic (Algorithm 3 in Appendix D.1) to find a projection matrix of rank $r = 200$. We then use Algorithm 2 to compute upper bounds on the $\infty \to 2$ operator norm of the projections matrices. Table 4 shows the values of the operator norms certified by our algorithm for the projection matrices that are found, along with the reconstruction/ projection error, expressed as a percentage. Notice the obtained subspaces have operator norm values significantly smaller than $\sqrt{n} \approx 34.641$.

## Footnotes

[3]Hence a fast algorithm that potentially finds a local optimum for the problem will not suffice for our purposes; we need an upper bound on the global optimum of (11).

[4]https://voice.mozilla.org/en/datasets