[Reviews · NeurIPS 2020]

Review 1

Summary and Contributions: The paper proposes to use low rank representations of the input to provide better guarantees for randomized smoothing techniques for adversarial robustness. The paper also proposes a way to extend the idea to the \ell_\infty setting, which is more challenging.

Strengths: Strength - I find it encouraging to see work that leverages internal structures in the data to improve randomized smoothing algorithms for probabilistic certificates of robustness. The work is well-motivated, easy-to-read and experimental results are easy to interpret (and I find them intuitively credible and is, I think, reproducible). Further, I like the formulation of the \infty \rightarrow 2 operator norm and the algorithm for certifying it. It is a clean intuitive formulation with good guarantees.

Weaknesses: Empirically, the benefit seems to be more prominent for higher values of perturbation radius and there seems to be a sacrifice in certified accuracy for smaller radii. I think it is natural to argue that sacrificing certified accuracy at smaller values of radii for better accuracy at larger perturbation radii is a slightly weird tradeoff to make. My guess would be that this is due to the information lost in the the low rank linear transformation. The low rank transformation done on the input space is both i) linear and ii) fixed i.e. not learnt during training, which introduces some bias (results in loss of information) into the training and thus results in the lesser accuracy for smaller perturbation radii where the information lost would have really helped. Thus, I would think it makes sense to look at/extend to non-linear learnt low rank representation learning algorithms like the one in Sanyal et. al, which also shows better robustness behaviour but perhaps more importantly robustness to gaussian perturbation. Thus, while I find the exploitation of internal structure in the data for randomized smoothing to be an useful direction for making randomized smoothing techniques more usable for l_\infty certification, I think the paper needs to improve further in not only using the structure present in the data but also the "intrinsic bias" present in these deep models. Some of these intrinsic biases help in obtaining a low rank representation space without this sacrifice in accuracy that occurs when the transformation is done in the input space. In particular, I think the authors should look at whether it is possible to extend the idea to representation space inside the architecture as opposed to just the input space as it might help to overcome the poor performance observed for smaller radii. There are already some hopeful results in the above mentioned references.

Correctness: I found no obvious mistakes in the theory or the empirical methodology.

Clarity: I found the paper very easy to read and interpret the results.

Relation to Prior Work: The paper does not discuss two very relevant work, which I believe are pertinent for reasons mentioned below - 1) Bafna et. al - which also discusses sparse fourier transforms as a method for inducing robustness for reasons very similar to the one discussed in the paper 2) Sanyal et. al. - which discusses learning low rank deep representation to induce robustness in deep neural networks.

Reproducibility: Yes

Additional Feedback: Some small comments are as follows: How is r chosen in Algorithm 1 ? Also please discuss the complexity of Algorithm 1. The no-go results mentioned in Line 46-48 are under some assumptions that are not necessarily a property of randomized smoothing methods. I think it would be better to be a bit more explicit about it. Bafna, M., Murtagh, J. and Vyas, N., 2018. Thwarting Adversarial Examples: An $ L_0 $-Robust Sparse Fourier Transform. In Advances in Neural Information Processing Systems (pp. 10075-10085). Sanyal, A., Kanade, V., Torr, P.H. and Dokania, P.K., 2018. Robustness via Deep Low-Rank Representations. ==== Additional Feedback===== I have appreciated the authors' feedback as it procides clarity on multiple topics like training complexity of the method, methodology for choice of r etc. Initially, I had appreciated the $\infty ->2$ operator presented in the paper as it seemed like a clever way of improving results on the $\ell_infty$ front and especially the connection via duality to $\infty -> 1$, and its connection to the Grothendieck problem and the algorithms presented therein. However, I read one of the cited papers [ACCV'19] in more detail after the review phase and I realized that the operator has actually been studied in quite detail along with its relation to the $\infty\rightarrow 1$ operator, the existing approximation algorithms etc. While the authors here have mentioned and cited ACCV'19 in this regard and did not claim novelty in these aspects, my review as well as some of the other reviewers' reviews, I have felt, had placed high importance on this. Regarding the advantage of the proposed approach against [SYL+19], I still think it is a bit concerning that the certified accuracy is lower for smaller radiuses and the difference gets larger for much smaller values of certified accuracy eg. after 35% accuracy in Fig 2b where the base accuracy of the randomized classified is 65%. I would argue that having an advantage for such small accuracies like 10,20,30% is pretty close to the trivial accuracy of a constant classifier. However, I do take the authors' point that in practice the radius of the adversarial perturbation need not be known in advance and hence, sacrificing a small accuracy for significant gain at higher radii is a good trade-off. At the same time, if the adversary knows that the framework does very well at large radii and slightly poorer at smaller radii then the adversary can always choose the smaller radii in order to gain small drops in certified accuracy without perturbing the image too much. Finally, I think the authors should talk about Bafna et al. which the authors have promised to discuss in the next updated version. Regarding Sanyal et. al., my point was not to compare with the robustness gained by non-linear dimensionality reduction approaches but whether smoothing smoothing can perhaps be done on this low rank feature representations. I take the authors' point that this violate most of the theory but I believe it can be an empirical extension of this work. I would also mention that I am not taking this last point into account in my review score as it is probably sufficiently different from the rest of the stuff in this paper. So, given all of this I am reluctant to increase my score and while I am not voting for rejection, I do not want to champion for acceptance.


Review 2

Summary and Contributions: This work exploits the fact that natural images often lie in a low dimensional subspace to provide certified robustness. Three main contributions are (1) they use PCA to project data in r-dimensional subspace and perform randomized smoothing in that space to obtain classifiers achieving certified robustness to l_2 perturbation. To avoid the argmax harness, they use cross-entropy loss over a soft classifier (a standard practice). (2) to translate guarantees from l_2 to l_inf, they find robust linear maps with small \inf->2 operator norm, and to do so, they propose a fast algorithm approximately solving a SDP relaxation to produce an upper bound. (3) they use the linear map found in (2) to train a network with certified robustness to l_inf perturbation.

Strengths: First of all, I really enjoyed reading this paper. According to me, the main strengths of this paper are: 1. The algorithm to compute upper-bound on the \inf->2 operator norm seems to be theoretically sound and might have use cases beyond just NN robustness literature. I find this to be very interesting. 2. transfer of robustness certificate from 2 to \inf norm is also interesting. 3. The work is very well motivated and the paper is written very well

Weaknesses: Overall, I have positive impression about the work. However, there are a few concerns (and questions) I have outlined below. I am more than happy to increase my score if the authors can provide satisfying answers to them: 1. Hyperparameters -- There are two additional hyperparameters here rank r and \lambda. I fear that the algorithm might be extremely sensitive to them. Please comment and also mention how do you justify cross validating these hyperparameters given that we already have additional hyperparameters in randomized smoothing. 2. Figure 1, the result is interesting. Did you try varying r (r was fixed to 200 in all your experiments) and checking how the certified accuracy changes? 3. To perform PCA, you need access to the entire dataset. How would you scale this to larger datasets such as ImageNet? 4. It is known that the features learned by NNs generally lie in a low-dimensional manifold. Is your approach, because of the projection at the input, going to push the rank of the features further down? This leads to another question, what happens if you directly enforce features to lie in a low-dimensional manifold. Can we still obtain better certified accuracy. Please comment. Minor: Line: 142, did you mean ‘blue’ instead of ‘yellow dotted line’?

Correctness: Yes, based on my quick evaluation, I find the method and the theory to be correct.

Clarity: Yes, the paper is very well written. The appendix did a very good job in filling most of the gaps which is absolutely fine.

Relation to Prior Work: Yes

Reproducibility: Yes

Additional Feedback: ==Post rebuttal== I appreciate the effort authors took to answer my questions, specifically the fact that they used reparameterization to learn the projection jointly, making the approach scalable to larger datasets. I am increasing my score as I am happy with the rebuttal. I would suggest authors to: 1. add these new results (joint training of projection matrix) in the updated draft, even if the results aren't great. 2. after reading the reviews of other reviewers, I guess the motivation for the above reparameterization came from the papers suggested by R1. I would suggest that the authors discuss those papers in their final draft if that's the case.


Review 3

Summary and Contributions: 1. This paper improved the certified L2 accuracy of randomized smoothing classifier by PCAing the images into a low rank representation space, where larger noises can be injected without affecting the natural accuracy empirically. 2. The transition from improved robust accuracy on L2 to Linf is natural but limited due to the inf->2 operator norm. The authors proposed a new algorithm to upper bound the inf->2 operator norm under PCA projection. The experiments showed the higher Linf robust accuracy in the DCT basis. After Rebuttal: the rebuttal answered the training complexity on PCA. As it is comparable to the adversarial training, I think it should be fine. I would keep my score.

Strengths: This paper is based on the certified robustness of randomized smoothing classifier. Improving robustness by projecting both images and noises into principle bases is intuitive. I haven't carefully check the algorithm 2 for upper bounding the inf->2 operator norm. From the experiment, it achieves better bounds compared to sqrt(n) in normal case.

Weaknesses: The PCA itself can hurt some natural accuracy as showed in Figure 2.

Correctness: It looks correct to me.

Clarity: This paper is neat and easy to follow. The core idea is clear.

Relation to Prior Work: The relation to previous papers is clearly discussed. The main training technique is borrowed from [1]. [1] Hadi Salman, Greg Yang, Jerry Li, Pengchuan Zhang, Huan Zhang, Ilya Razen- shteyn, and Sebastien Bubeck. Provably robust deep learning via adversarially trained smoothed classifiers.

Reproducibility: Yes

Additional Feedback: I have some questions about the PCA step. 1. How about the training computational cost compared to [1] since your algorithm requires PCA to every image and channel? 2. Can other compression models (e.g., autoencoder) be used here instead of PCA?


Review 4

Summary and Contributions: This paper studies certificated L_2 and L_inf robustness. Authors give a new smoothed classifier which can achieve certificated L_2 robustness guarantee. Then, they show how to transfer it to the L_inf case. In addition, they show that if the input has certain properties, there is a better way to achieve certificated L_inf robustness.

Strengths: The authors study the certificated robustness when the input falls in a low dimensional subspace. The L_2 algorithm is simple and easy to implement. The l_inf algorithm is much involved. They reduce the problem to finding a projection matrix with small L_inf->L_2 norm. Their algorithm is based on MWU for approximately solving SDP which looks interesting. In their experiment, they evaluated their theoretical L_2 and L_inf algorithms, and they also show that they can use the projection obtained by DCT basis and get good results.

Weaknesses: I feel the technical contribution is somewhat weak. For the L_2 algorithm, once the input is assumed in a low dimensional space, it is very natural to use the L_2 projection matrix to restrict the perturbation in the subspace. Then the analysis follows from the previous work. For the L_inf part, I think the reduction to finding a projection matrix P with small L_inf->L_2 norm is good. But the proposed method (SDP solver based) is somewhat straightforward. Since it only needs a course upper bound of the norm, it is not clear whether there is a more efficient way to obtaining the projection matrix. DCT based approach is a potential choice, however, authors do not have theoretical analysis for that.

Correctness: I did not find obvious issues in their proof or the experiments.

Clarity: The paper is well-written in general.

Relation to Prior Work: In Appendix, they have a detailed comparison with previous papers.

Reproducibility: Yes

Additional Feedback: - It would be good to compare the running time between DCT based algorithm and SDP based algorithm. - If MWU SDP based algorithm is much slower than DCT based algorithm, I will appreciate if authors can give a more efficient theoretical algorithm for finding projection P with small L_inf->L_2 norm. If it is hard for general inputs, I think it is good to analyze some special inputs. For example, under which kind of theoretical assumption of inputs, DCT based algorithm has theoretical guarantee. Post rebuttal=========================== Authors addressed some of my confusions. So I would still keep my positive attitude for accepting the paper.

[Author Response · NeurIPS 2020]

We thank all the reviewers for their insightful comments and suggestions. We will update the final version accordingly.

**R#1.** We thank the reviewer for pointing us to the two relevant references (we will discuss them in the final version),
that use low-rank representations to get robustness to specific empirical attacks like PGD. We remark that in our work
low rank representations are used in a different way. To provide the improved *certified* robustness guarantees, we take
advantage of good low-rank representations in extending the randomized smoothing approach appropriately.

*(a) On using non-linear projections:* We agree that introducing a non-linear dimensionality operation may lead to
classifiers that are more robust against PGD/FGSM style attacks, as observed in Sanyal et al. Our setting is quite
different since our aim is to produce classifiers with *certified* accuracy guarantees. Introducing a projection operation at
an intermediate layer completely breaks down the theoretical analysis of certified accuracy. Hence, we are restricted
to using linear projections in order to provide provable guarantees for our classifier. We believe that the reviewer's
suggestion of using/analyzing non-linear projections is an excellent direction for future work.

*(b) On training the linear projection vs. using a fixed projection:* We agree that in certain cases such as text data where
the input representation is not fixed, training the linear projection along with the network could be beneficial and in
fact necessary. We are currently exploring this direction. For vision datasets that we used in the paper, we indeed had
experimental results with simultaneously training the projection with the network parameters. The results we obtained
were similar to using a fixed projection and we did not see any significant advantage. We chose to present the simpler
approach to convey the core idea clearly. We will include these results in the final version.

*(c) Choice of $r$:* We plot the PCA reconstruction error as a function of $r$ and choose a value of $r$ in a certain range where
the error is not too high (less than $3\%$). See Fig. 5. There are multiple choices of $r$ that work equally well.

*(d) Training complexity:* The complexity of Algorithm 1 is comparable to the complexity of training a smoothed
classifier as in the work of [SYL$^+$19]. The PCA step incurs a one time preprocessing cost and the projection step at
the beginning simply corresponds to adding a linear layer to an existing ResNet architecture. As an example, on the
CIFAR-10 dataset, for $\epsilon = 0.25$, training the classifier of [SYL$^+$19] takes on average 21.27 seconds per epoch, whereas
Algorithm 1 takes 21.29 seconds per epoch on average. The same behavior holds across different parameter settings.

*(e) On trading off certified accuracy vs. natural accuracy:* Notice that for most values of $\epsilon$, as Fig. 2 shows, we suffer
almost no loss in accuracy at small radii. Additionally, as we state in Line 141 (Page 4), for a large range of values for
the robustness radius, our method gets much better natural accuracy for a desired robust accuracy and radius. In practice,
we may not know the radius of adversarial perturbation (and the ideal choice of $\epsilon$) beforehand, hence sacrificing a small
amount of accuracy at small radii for a significant gain at higher radii is a desirable tradeoff.

**R#2.** In Line 142, thanks for catching the typo: yes we meant blue instead of yellow. We now address the other points.

*(a) Scaling PCA to large datasets:* One can perform PCA on a smaller random sample to get the projection. Alternately,
one can also train the network and the projection operator simultaneously. Formally, reparameterize $\Pi = UU^T$ and
augment the loss function with two terms: 1) Reconstruction error on the **mini batch** namely, $\|x - Ux\|^2$; and 2)
$\|U^T U - I\|_F^2$ to encourage $U$ to be orthonormal. Our experiments with this approach on CIFAR-10/100 show results
similar to those reported in the paper. Further, this approach naturally scales to large datasets.

*(b) Certified accuracy vs. $r$ & hyperparameters:* Changing $r$ by small amounts had negligible effect on the certified
accuracy (we tried values of $r$ close to 200). The smaller we can keep $r$, the more robustness we can achieve. As
mentioned in our response to R#1 (1c), the reconstruction error (see Fig. 5) dictated our choice of $r = 200$. This choice
made apriori to the training stage works well across different settings, suggesting that choice of hyperparameters can be
decoupled. However, we did not run extensive experiments for end to end training for many different values of $r$. The
performance of our algorithm is smooth in $\lambda$ (see Fig. 1). Furthermore, in all our experiments $\lambda = 0.5$ worked very well
pointing to the fact that there are generic settings of $\lambda$ that one can often use. We will clarify more in the final version.

*(c) On feature dimensionality:* We do not know if linear projection forces the intermediate representations to be even
lower dimensional. Forcing intermediate representations to be low dimensional may certainly lead to empirical benefits.
However, it seems challenging to use this to get certified accuracy guarantees.

**R#3.** Please see responses to R#1 for the effect of PCA on natural accuracy (1a) and training cost (1d). Using
autoencoders is an interesting idea. Currently we do not know how to use them to get certified accuracy guarantees.

**R#4.** Regarding *new SDP algorithm* we respectfully disagree with the opinion that the approach is straightforward.
Please note two main contributions: (i) the algorithm is practical and novel within the broad MWU-framework (with an
important change to the weight update) , (ii) the accompanied theoretical guarantee gives a significant improvement
over the current SOTA, for an even broader class of general quadratic programs (as discussed in Secs. 3, A, and D). As
experimentally shown in Sec. F, the algorithm is much faster than existing solvers, and may be of independent interest.

*(a) MWU vs DCT:* We are somewhat confused by the reviewer's question about comparing MWU and DCT. These are
not two competing approaches. For certified $\ell_\infty$ guarantees in the DCT domain, these are used together. The MWU
based algorithm is used as a subroutine to find a good robust low-rank representation in the DCT domain (the MWU
based algorithm is crucial for obtaining certified guarantees).

[Meta-Review · NeurIPS 2020]

The authors present a nice way to boost the effectiveness of gaussian smoothing. The proposed method is unlikely to work on complex datasets like ImageNet, but the method is clever and gives a big enough boost on toy problems to justify publication.